# Coresets for Decision Trees of Signals

**Ibrahim Jubran**
Department of Computer Science
University of Haifa, Israel
ibrahim.jub@gmail.com

**Ernesto Evgeniy Sanches Shayda**
Department of Computer Science
University of Haifa, Israel
ernestosanches@gmail.com

**Ilan Newman**
Department of Computer Science
University of Haifa, Israel
ilan@cs.haifa.ac.il

**Dan Feldman**
Department of Computer Science
University of Haifa, Israel
dannyf.post@gmail.com

## Abstract

A $k$-decision tree $t$ (or $k$-tree) is a recursive partition of a matrix (2D-signal) into $k \geq 1$ block matrices (axis-parallel rectangles, leaves) where each rectangle is assigned a real label. Its regression or classification loss to a given matrix $D$ of $N$ entries (labels) is the sum of squared differences over every label in $D$ and its assigned label by $t$. Given an error parameter $\varepsilon \in (0, 1)$, a $(k, \varepsilon)$-coreset $C$ of $D$ is a small summarization that provably approximates this loss to *every* such tree, up to a multiplicative factor of $1 \pm \varepsilon$. In particular, the optimal $k$-tree of $C$ is a $(1 + \varepsilon)$-approximation to the optimal $k$-tree of $D$.

We provide the first algorithm that outputs such a $(k, \varepsilon)$-coreset for *every* such matrix $D$. The size $|C|$ of the coreset is polynomial in $k \log(N)/\varepsilon$, and its construction takes $O(Nk)$ time. This is by forging a link between decision trees from machine learning – to partition trees in computational geometry.

Experimental results on `sklearn` and `lightGBM` show that applying our coresets on real-world data-sets boosts the computation time of random forests and their parameter tuning by up to x10, while keeping similar accuracy. Full open source code is provided.

## 1   Introduction

Decision trees are one of the most common algorithms used in machine learning today, both in the academy and industry, for classification and regression problems [52]. Informally, a decision tree is a recursive binary partition of the input feature space into hyper-rectangles, where each such hyper-rectangle is assigned a label. If the labels are given from a discrete set, the trees are usually called *classification trees*, and otherwise they are usually called *regression trees*. Variants include non-binary partitions and forests [29].

**Why decision trees?** Advantages of decision trees, especially compared to deep learning, include: **(i)** Interpretability. They are among the most popular algorithms for interpretable (transparent) machine learning [31]. **(ii)** Usually require small memory space, which also implies fast classification time. **(iii)** Accuracy. Decision trees are considered as one of the few competitors of deep networks. In competitions, such as the ones in Kaggle [36], they are one of the favorite classifiers [10], especially on small or traditional tabular data. **(iv)** May learn from small training data.

**The goal** is usually to compute the optimal $k$-tree $t^*$ for a given dataset $D$ and a given number $k$ of leaves, according to some given loss function. In practice, researchers usually use ensemble of trees called forests, e.g., a Random Forest [12], which are usually learned from different subsets of the

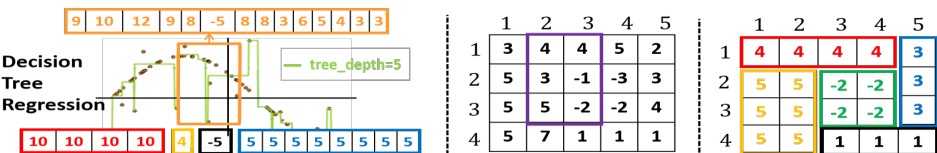

Figure 1: **(Left):** A one dimensional signal (orange points) and its segmentation into 25 "smooth" segments / leaves (green lines). Image taken from Section 1.10 ("Decision Trees") of the sklearn's User Guide [56]. The vector $v$ on top represents a subset of the signal's values. The bottom vectors represent a 4-segmentation of $v$, similar to the horizontal green line segments. Each segment contains the average value of its corresponding segment from $v$. **(Middle):** A matrix that represents the $4 \times 5$ signal $D = \{((1,1),3),((1,2),4),((1,3),5),\cdots\}$ (in black) and a $3 \times 2$ matrix that represents a $3 \times 2$ sub-signal $B = \{((1,2),4),((1,3),4),((2,2),3),\cdots\}$ (in purple). **(Right):** A matrix that represents a 5-segmentation $s$ of $A = [4] \times [5]$; see Definition 1. Since $s$ is a 5-segmentation, it partitions the $[4] \times [5]$ matrix into 5 distinct block matrices $B_1$ (red), $B_2$ (blue), $B_3$ (yellow), $B_4$ (green), and $B_5$ (black) such that $s$ assigns the same number for all the entries in the same block. The SSE fitting loss $\ell(D, s)$ is the sum of squared differences over every entry in the left matrix to its corresponding entry on the right matrix. Also, there is no $k$-tree that can obtain the same partition.

training set. The final classification is then based on a combination rule, such as majority or average vote. Since both the training and classification of each tree are computed independently and possibly in parallel, we focus on the construction of a single tree.

A **_dataset_** $D$ in this paper is a set $D = \{(x_1, y_1), \cdots, (x_N, y_N)\} \subseteq A \times \mathbb{R}$ of pairs, where $A$ is the *feature space*. Each pair $(x, y) \in D$ consists of a database record (vector / sample) $x \in A$ and its real label $y \in \mathbb{R}$. As common, we assume that non-real features, such as categorical features, are converted to real numbers; see e.g. [27]. For example, in common classification problems $A = \mathbb{R}^d$ for some $d \geq 1$ and $y \in \{0, 1\}$ is a binary number. The resulting model may be used for prediction on another test dataset, completion of missing values, or efficient storage of the original dataset by replacing the label $y$ of each pair $(x, y) \in D$ with the label $t(x)$ that was assigned to it by the tree $t$. The last technique is used e.g. in the MPEG4 encoder [46], where decision trees of a specific structure (quad-trees) are used to compress an image $D$ that consists of pixel-grayscale pairs $(x, y)$.

**Challenges.** The motivation for this paper originated from the following challenges:
**(i) Sub-optimality.** Hardness of decision tree optimization is both a theoretical and practical obstacle [31]. It is NP-hard to compute the optimal $k$-tree, or its approximation, when the number $k$ is not fixed [14, 39]. There were several attempts to improve the optimality of decision tree algorithms, from binary-split decision trees as in [6, 8], in a line of work of e.g. [9, 61]. Nevertheless, greedy implementations e.g., CART [40] and C4.5 [51] have remained the dominant methods in practice.
**(ii) Computation time.** Due to this lack of optimality, finding a decision tree that provides a good accuracy usually requires many runs, since each of them returns only a local minimum that might be arbitrarily far from the global optimum. The final model usually consists of a forest containing many trees. Popular forest implementations include the famous Sklearn, XGBoost, LightGBM, and CatBoost libraries [49, 15, 37, 19], which all utilize (as default) an ensemble of at least 100 trees. Moreover, there is a list of dozen parameters to calibrate including: number of trees, depth of each tree, pruning/splitting strategies on each tree and between them, and many others. To this end, the running time for obtaining reasonable results even on moderate size datasets might be impractical.
**(iii) Scalability.** Existing techniques tend not to scale to realistically-sized problems unless simplified to trees of a specific form as stated in [31].
**(iv) Streaming, parallel, and dynamic updates.** The common algorithms mentioned above do not support continuous learning or updating of the modeled tree when an input sample is either added or removed from the dataset, e.g., when the dataset does not fit into memory or arrives on-the-fly. Similarly, we do not know techniques to train a single tree in parallel on multiple machines.

## 1.1 Coresets

"Coresets are one of the central methods to facilitate the analysis of large data sets." [47]. Informally, for an input dataset $D$, a set $T$ of models, an approximation error $\varepsilon \in (0, 1)$, and a loss function $\ell$, a coreset $C$ is a data structure that approximates the loss $\ell(D, t)$ for every model $t \in T$, up to

a multiplicative factor of $1 \pm \varepsilon$, in time that depends only on $|C|$. Hence, ideally, $C$ is also much smaller than ther original input $D$.

**Why coresets?** The main motivation for constructing a coreset is to compute the optimal model or its approximation, much faster, while sacrificing little accuracy. Furthermore, a coreset for a family of classifiers is many times a "silver bullet" that provides a unified solution to all Challenges (i)-(iv) above. Combining the two main coreset properties: merge and reduce [32, 7, 26, 1], which are usually satisfied, with the fact that a coreset approximates every model, and not just the optimal model, enables it to support streaming and distributed data [11, 41], parallel computation [21], handle constrained versions of the problem [25], model compression [20], parameter tuning [43] and more.

**Coreset construction techniques.** There are many different techniques for coreset construction, ranging from loss-less to lossy, from deterministic to randomized, and from greedy to non-greedy constructions. Examples include accurate coresets [33] and non-accurate coresets [23] via computational geometry, random sampling-based coresets [22, 17, 44], and greedy deterministic coresets via the Frank-Wolfe algorithm [16]. Recently, many works focus on developing frameworks for general families of loss functions, e.g., [22, 57]. We refer the interested reader to the surveys [2, 1, 50, 11, 4, 21] with references therein.

**Practical usage.** Since a coreset is not just another solver that competes with existing solutions, but a data structure for approximating any given model in the family, we can apply *existing* approximation algorithms or heuristics on the coreset to obtain similar results compared to the original (full) data. Since the coreset is small, we may run these heuristics multiple times, or apply the hyperparameter tuning using the coreset [43], thus reducing the computational burden by orders of magnitude.

**Main challenges: (i)** No coreset. Unfortunately, it is not clear at all that a small coreset exists for a given family of models. In fact, we can conclude from [54] that a coreset for decision trees does not exist in general; see details below. In this case, we can either give up on the coreset paradigm and develop a new solver, or add assumption on the input dataset, instead of targeting every possible dataset that may be very artificial and unrealistic, as the counter example in [54]. In this paper, we choose the latter option. **(ii)** Unlike, say, uniform sampling, every problem formulation requires a different coreset construction, which may take years of research to design.

## 1.2 First coreset for decision trees and their generalization

In this paper, we tackle a generalized and more complex set of models than decision trees, where, rather than a recursive binary partition, we allow the input feature space $\mathbb{R}^d$ to be partitioned into any $k$ disjoint axis-parallel hyper-rectangles; see Fig. 1. This generalization is essential in order to support future non-recursive and not necessarily binary classification models, e.g., ID3 and C4.5 [30, 51]. To our knowledge, this is the first coreset with provable guarantees for constructing decision trees.

**Definition 1** ($k$-segmentation). *For an integer $d \geq 1$ that denotes the dimension of the* feature space *$A = \mathbb{R}^d$, and an integer $k \geq 1$ that denotes the size of the partition (number of leaves), a function $s : A \to \mathbb{R}$ is a $k$-segmentation if there is a partition $\mathcal{B} = \{B_1, \cdots, B_k\}$ of $A$ into $k$ disjoint axis-parallel hyper-rectangles (blocks), such that $|\{s(b) \mid b \in B\}| = 1$ for every block $B \in \mathcal{B}$, i.e., $s$ assigns a unique value for all the entries in each of its $k$ rectangles; see Fig. 1. We define the union over all possible such $k$-segmentations by $\mathrm{SEG}_{(k,d)}$.*

We now define our loss function, and an optimal $k$-segmentation model over some set $A \subseteq \mathbb{R}^d$.

**Definition 2** (Loss function). *For a dataset $D = \{(x_1, y_1), \cdots, (x_N, y_N)\} \subseteq A \times \mathbb{R}$, an integer $k \geq 1$, and a $k$-segmentation $s \in \mathrm{SEG}_{(k,d)}$, we define the* sum of squared error (SSE) *loss*

$$\ell(D, s) := \sum_{(x,y) \in D} (s(x) - y)^2$$

*as the loss of fitting $s$ to $D$. A $k$-segmentation $s^*$ is an* optimal $k$-segmentation *of $D$ if it minimizes $\ell(D, s)$ over every $k$-segmentation $s \in \mathrm{SEG}_{(k,d)}$ i.e., $s^* \in \arg\min_{s \in \mathrm{SEG}_{(k,d)}} \ell(D, s)$. The optimal SSE loss is denoted by $\mathrm{opt}_k(D) := \ell(D, s^*)$.*

For example, the optimal 1-segmentation $s^*$ of $D$ is the constant function $s^* \equiv \frac{1}{|D|} \sum_{(x,y) \in D} y$ since the mean of a set of numbers minimizes the sum of squared distances to the elements of the set. Also, $\mathrm{opt}_{|D|}(D) = 0$ for every dataset $D$.

We are now ready to formally define a coreset for the $k$-segmentation problem (and $k$-decision trees of at most $k$ leaves, in particular).

**Definition 3** (Coreset). *Let $D = \{(x_1, y_1), \cdots, (x_n, y_n)\} \subseteq A \times \mathbb{R}$ be an input dataset. Let $k \geq 1$ be an integer and $\varepsilon \in (0, 1)$ be the desired approximation error. A $(k, \varepsilon)$-coreset for $D$ is a data structure $(C, u)$ where $C \subseteq A \times \mathbb{R}$ is an ordered set, and $u : C \to [0, \infty)$ is called a* weight function*, such that $(C, u)$ suffices to approximate the loss $\ell(D, s)$ of the original dataset $D$, up to a multiplicative factor of $1 \pm \varepsilon$, in time that depends only on $|C|$ and $k$, for any $k$-segmentation $s$.*

**Practical usage.** As defined above and discussed in Section 1.1, a coreset approximates every model in our set of models $\mathrm{SEG}_{(k,d)}$. Hence, a coreset for decision trees is clearly also a coreset for forests with an appropriate tuning for $k$, since every tree in the forest is approximated independently by the coreset. We expect that applying existing heuristics (not necessarily with provable guarantees) such as sklearn [49] or LightGBM [37] on the coreset, would yield similar results compared to the original data. Indeed, our experimental results in Section 5 validate those claims.

**No coreset for general datasets.** Unfortunately, even for the case of $k = 4$ and $A \subseteq \mathbb{R}$, i.e., when the input is simply a one dimensional dataset $D = \{(x_1, y_1), \cdots, (x_n, y_n)\}$ where $x_1, \cdots, x_n$ are real numbers, and the labels $y_1, \cdots, y_n \in \{0, 1\}$ have only binary values, it is easy to construct datasets which have provably no $k$-segmentation (or even $k$-tree) coreset of size smaller than $n$; see e.g. [54]. Hence, there is no non-trivial decision tree coreset for general datasets of $n$ vectors in any dimension. However, as we prove in the rest of the paper, a coreset does exist for datasets where the input is a matrix, i.e., a discrete signal where every coordinate in the domain is assigned a label (value), rather than a random set of $n$ vectors.

**The first coreset for $n \times m$-signals.** To overcome the above problem, while still obtaining a small coreset, we assume a discretization of the dataset so that every coordinate has a label. We also assume, mainly for simplicity and lack of space, that the input feature space is $A = [n] \times [m] \subseteq \mathbb{R}^2$. That is, the input can be represented by an $n \times m$ matrix. The output coreset may contain fraction of entries, as in Fig. 3, which is called an $n \times m$ *signal*; see Section 1.5. Our assumption on the input data seems to be the weakest assumption that can enable us to have a provably small coreset for any input. Furthermore, it seems natural for e.g. images, matrices, or any input data from sensors (such as GPS) that has a value in every cell or continuous in some other sense.

**Previous work.** The prior works [54, 24, 62], which only handle the case of segmenting a 1-dimensional signal, use relaxations similar to our relaxation above to obtain a coreset of size $O(k/\varepsilon^2)$. However, our results apply easily for the case of vectors (1-dimensional signals) as in [54] and generalize for tensors if $d \geq 3$. We also give further applications, and provide extensive experiments with popular state of the art software.

A special case for $d = 2$ includes image compression, where quadtrees are usually used in e.g. MPEG4 to replace the image by smooth blocks of different sizes [55], or for completion of missing values [58] Using dynamic programming, it is easy to compute the optimal tree of a 2D-signal $D$ in $O(k^2 n^5)$ time [5], which is impractical even for small datasets, *unless applied on a small coreset of $D$*. However we do not know of any such coreset construction, for $d \geq 2$, with provable guarantees on its size. To this end, the following questions are the motivation for this paper: **(i):** Is there a small coreset for any $n \times m$ signal (e.g. of sub-linear size)? **(ii):** If so, can it be computed efficiently? **(iii):** Can it be used on real-world datasets to boost the performance of existing random forest implementations?

**Extensions.** For simplicity, we focus on the classic sum of squared distances (SSE) or the risk minimization model [59]. However, our suggested techniques mainly assume that a coreset for the case $k = 1$ is known, which is trivial for SSE, but exists for many other loss functions e.g., non-squared distances; see Section 6.

## 1.3 Our Contribution

For any given error parameter $\varepsilon \in (0, 1)$, and an integer $k \geq 1$, this paper answers affirmatively the above three questions. More formally, in this paper we provide:

**(i):** A proof that *every* $n \times m$ signal $D$ has a $(k, \varepsilon)$-coreset $(C, u)$ of size $|C|$ polynomial in $k \log(nm)/\varepsilon$. To our knowledge, this is the first coreset for decision trees whose size is smaller than the input; see Theorem 8. Due to lack of space, our full proofs are given in the appendix.

**(ii):** A novel coreset construction algorithm that outputs such a coreset $(C, u)$ with the above guarantees, for every given input signal $D$. Its running time is $O(nmk)$, i.e., linear in the input size

$|D|$. Unlike common coreset constructions, our algorithm is deterministic; see Algorithm 3.
**(iii):** AutoML for decision trees: since the suggested coreset approximates *every* tree of *at most $k$* leaves, we may use the same coreset for hyperparameter tuning. We demonstrate this in Section 5, by calibrating the parameter $k$ using only the coreset, as compared to using the original (big) data.
**(iv):** Experimental results that apply modern solvers, such as the sklearn and LightGBM libraries, on this coreset for real-world public datasets. We show that our coreset can help boost the computation time of the above forest implementations and their parameter tuning by up to x10, while keeping similar accuracy; see Section 5.
**(v):** Open source code for our algorithms [35]. We expect that it will be used by both the academic and industry communities, not only to improve the running time of existing projects, but also to extend the algorithms and experimental results to other libraries and cost functions; see Section 6.

## 1.4 Novel technique: partition trees meet decision trees

In a seminal paper [28] during the 80's of the previous century, Haussler and Welz introduced the importance of VC-dimension by Vapnik–Chervonenkis [60]. Their main application was partition trees for answering range queries.

Informally, a *partition tree* of a given set of points on the plane is the result of computing recursively a *simplicial partition* which is defined as follows. For a set $D$ of $N$ points on the plane, a $(1/\varepsilon)$-*simplicial partition* is the partition of $D$ into $O(1/\varepsilon)$ subsets, such that: (i) each subset has at most $2\varepsilon N$ points, and (ii) Every line in the plane intersects the convex hull of at most $\sqrt{1/\varepsilon}$ sets. Answering range queries of the form "how many points in $D$ are in a given rectangular" in sub-linear time, using partition trees, is straightforward: We can sum in $O(1/\varepsilon)$ time the number of points in the subsets of the above simplicial partition that are not intersected by the query rectangular. We then continue recursively to count the points on each of the $\sqrt{1/\varepsilon}$ intersected sets.

In other words, the main idea behind the above work is to partition the input into a (relatively small) number of subsets, each containing a fraction of the input, such that each query (in this case, a rectangular shape) might intersect only a small fraction of those subsets. Such a partition is termed a simplicial partition. The number of points contained in non-intersected subsets can be easily computed, while the sum of points in intersected subsets require a more involved solution. The novelty in that work is how to achieve such a partition of the input.

Our paper closes a loop in the sense that it forges links between decision trees in machine learning – to partition trees from computational geometry. We aim to generalize the above technique from covering problems to regression and classification problems. This is by devising an algorithm which achieves the above requirements, but where the query is a decision tree (and not a rectangular shape), and the cost function is the sum of squared distances to the query and not the number of points.

More precisely, We partition the input dataset $D$ into a relatively small number of subsets, such that every possible decision tree (query) intersects at most few of these subsets. We then independently compress every subset via another novel algorithm such that the cost (sum of squared distances, in this case) of points contained in non-intersected subsets can be easily and accurately estimated, while the cost of points in intersected subsets can be provably approximated via a more involved calculation. This is very different from existing coreset techniques that are sampling-based [38], or utilize convex optimization greedy algorithms [16]. Our main challenge was to define and design such a "simplicial partition for sum of squared distances", and the coreset to be computed for each subset in this partition.

## 1.5 Preliminaries

In this section we define the notation that will be used in the next sections.

Let $n \geq 1$ and denote $[n] = \{1, \cdots, n\}$. An $n$-signal is a set $\{(x, f(x)) \mid x \in [n]\}$ that is defined by a function $f : [n] \to \mathbb{R}$ (known as the graph of $f$). For an additional integer $m \geq 1$, an $n \times m$ signal $D = \{(x, g(x)) \mid x \in [n] \times [m]\}$ is the set that corresponds to a function $g : [n] \times [m] \to \mathbb{R}$. That is, $D$ represents an $n \times m$ real matrix whose size is $|D| = N = nm$. For integers $i_1, i_2, j_1, j_2$ such that $1 \leq i_1 \leq i_2 \leq n$ and $1 \leq j_1 \leq j_2 \leq m$, an $n \times m$ sub-signal is the set $B = \{(x, g(x)) \mid x \in \{i_1, \cdots, i_2\} \times \{j_1, \cdots, j_2\}\} \subseteq D$; see Fig. 1. A sub-signal $B$ is called a *row* (respectively, *column*) if $i_1 = i_2$ (respectively, $j_1 = j_2$). For a sub-signal $B$, we

denote by $B^T = \{((j, i), y) \mid ((i, j), y) \in B\}$ the *transposed sub-signal*. A $k$-segmentation $s$ is said to *intersect* an $n \times m$ sub-signal $B$ if $s$ assigns at least two distinct values to the entries of $B$, i.e., $|\{s(x) \mid (x, y) \in B\}| \geq 2$. Furthermore, by definition, a $k$-segmentation $s$ induces a partition of an $n \times m$ sub-signal $B$ into at most $k$ $n \times m$ sub-signals. For two functions $f, g : \mathbb{R} \to \mathbb{R}$ we use the big $O$ notation $f(x) \in O(g(x))$, thinking of $O(g(x))$ as the class of all functions $h(x)$ such that $|h(x)| \leq c|g(x)|$ for every $x > x_0$, for some constants $c$ and $x_0$. Lastly, we denote $\mathrm{SEG}_k := \mathrm{SEG}_{(k,2)}$ for brevity.

**Paper organization.** Section 2 provides a rough approximation to the $k$-segmentation problem. Section 3 provides an algorithm for computing a simplicial partition for the $k$-segmentation problem. Each region in this partition will be then compressed individually in Section 4 to obtain our desired coreset. Experimental results and discussions are given in Section 5, and a conclusion in Section 6.

## 2   Bi-criteria Approximation

A coreset construction usually requires some rough approximation to the optimal solution as its input. Unfortunately, we do not know how to *efficiently* compute even a constant factor approximation to the optimal $k$-segmentation problem in Definition 2, as explained in Section 1. Instead, we provide an $(\alpha, \beta)_k$ or bi-criteria approximation [22], where the approximation is with respect to a pair of parameters: the number of segments in the partition may be up to $\beta k$ instead of $k$, and the loss may be up to $\alpha \cdot \mathrm{opt}_k(D)$ instead of $\mathrm{opt}_k(D)$.

**Definition 4** $((\alpha, \beta)_k$-approximation.). *Let $D$ be an $n \times m$ sub-signal, $k \geq 1$ be an integer and let $\alpha, \beta > 1$. A function $s : [n] \times [m] \to \mathbb{R}$ is an $(\alpha, \beta)_k$-approximation of $D$, if $s$ is a $\beta k$-segmentation whose fitting loss to $D$ is at most $\alpha$ times the loss of the optimal $k$-segmentation of $D$, i.e., $s \in \mathrm{SEG}_{(\beta k)}$ and $\ell(D, s) = \sum_{(x,y) \in D} (s(x) - y)^2 \leq \alpha \cdot \mathrm{opt}_k(D)$.*

We now describe an algorithm that computes such an approximation, in time only linear in the input's size $|D| = nm$. The following lemma gives the formal statement. A suggested implementation for the algorithm is given in the appendix, as well as the full proof of the lemma; see Section B.

**Lemma 5.** *Let $D = \{(x_1, y_1), \cdots, (x_{nm}, y_{nm})\}$ be an $n \times m$ sub-signal and $k \geq 1$ be an integer. Then, there is an algorithm that can compute, in $O(knm)$ time, an $(\alpha, \beta)_k$-approximation for $D$, where $\alpha \in O(k \log(nm))$ and $\beta \in k^{O(1)} \log^2(nm)$.*

**Overview of the bicriteria algorithm from Lemma 5:**   The algorithm is iterative and works as follows. At the $i$th iteration, we find a collection $\mathcal{B}_i$ of at most $t$ disjoint sub-signals in $D_i$ (where $D_0 = D$ is the input), for which: (i) $\sum_{B \in \mathcal{B}_i} \mathrm{opt}_1(B) \leq \mathrm{opt}_k(D_i) \leq \mathrm{opt}_k(D)$, and (ii) $\cup_{B \in \mathcal{B}_i} B$ has size $|\cup_{B \in \mathcal{B}_i} B| \geq |D_i|/c$ for some parameter $c$ that depends on $k$, i.e., those sub-signals contain at least a $1/c$ fraction of $D_i$. We then define $D_{i+1} = D_i \setminus \cup_{B \in \mathcal{B}_i} B$. After repeating this for at most $\psi \in O(c \log(nm))$ iterations, we end up covering all entries of $D$ with sub-signals where the overall loss of the sub-signals in each iteration is at most $\mathrm{opt}_k(D)$. This defines a partition of $D$ into a collection of at most $t\psi$ disjoint sets $\mathcal{B}'$, which, in turn, define a set of at most $(t\psi)^2$ distinct sub-signals. The output is now simply the function $s$ that assigns, for every $B \in \mathcal{B}'$ and $b \in B$, the mean value $s(b) = \frac{1}{|B|} \sum_{((i,j),y) \in B} y$ of $B$. See Pseudo-code in Algorithm 4 at the appendix.

## 3   Balanced Partition

In this section we present Algorithm 2, which computes a partition similar to the simplicial partition described in Section 1.4; It computes, in $O(|D|)$ time, a partition $\mathcal{B}$ of the input $D$ that satisfies the following properties: (i) $|\mathcal{B}|$ depends on $k/\varepsilon$ but independent of $|D|$, (ii) the loss $\mathrm{opt}_1(B)$ of every $B \in \mathcal{B}$ is small, and (iii) every $k$-segmentation $s$ intersects only few sub-signals $B \in \mathcal{B}$; see Definition 6, Fig. 2, and Lemma 7. A full proof is given at the appendix; see Section C.

**Definition 6** (Balanced Partition). *Let $D$ be an $n \times m$ signal, $k \geq 1$ be an integer, and $c_1, c_2, c_3 > 0$. A $(c_1, c_2, c_3)_k$-balanced partition of $D$ is a partition $\mathcal{B}$ of $D$ such that: (i) $\mathcal{B}$ contains $|\mathcal{B}| \leq c_1 n \times m$ sub-signals, (ii) $\mathrm{opt}_1(B) \leq c_2$ for every $B \in \mathcal{B}$, and (iii) every $k$-segmentation $\hat{s}$ intersects at most $c_3$ sub-signals $B \in \mathcal{B}$ (i.e., assigns more than one unique number to those sub-signals).*

**Lemma 7.** *Let $D$ be an $n \times m$ signal, $k \geq 1$ be an integer, $\varepsilon \in (0, 1/4)$ be an error parameter, and $s : [n] \times [m] \to \mathbb{R}$ be an $(\alpha, \beta)_k$-approximation of $D$, where $\alpha, \beta > 1$. Define $\sigma := \frac{\ell(D,s)}{\alpha}$*

Figure 2: **(Left):** A $6 \times 5$ signal $D$ consisting of 6 rows $R_1, \cdots, R_6$. **(Middle):** A step by step illustration of the call to $\mathcal{B} := \text{PARTITION}(D, 1/4, 64)$ which partitions $D$ into $|\mathcal{B}| = 13$ sub-signals (sub-matrices) as follows. (1) $\mathcal{B}' := \text{SLICEPARTITION}(\{R_1\}, 4)$ (top green row). (2) $\mathcal{B}' := \text{SLICEPARTITION}(R_1 \cup R_2, 4)$ (middle green matrix), and so on as long as the output contains at most $|\mathcal{B}'| \leq 1/\gamma = 4$ sub-signals (as in the bottom green matrix (3)). We then append $B'$ to the output $\mathcal{B}$, and repeat with the remaining $\{R_4, R_5, R_6\}$. (4) $\mathcal{B}' := \text{SLICEPARTITION}(\{R_4\}, 4)$ which already returns $|\mathcal{B}'| = 5 > 1/\gamma$ signals (yellow matrix). We append them to $\mathcal{B}$ and repeat. **(Right):** The final partition $\mathcal{B}$, where $\text{opt}_1(B) \leq \gamma^2 \sigma = 1/4^2 \cdot 64 = 4$ for every $B \in \mathcal{B}$.

and $\gamma := \frac{\varepsilon^2}{\beta k}$. Let $\mathcal{B}$ be an output of a call to $\text{PARTITION}(D, \gamma, \sigma)$; see Algorithm 2. Then $\mathcal{B}$ an $\left( O\left(\frac{\alpha}{\gamma^2}\right), \gamma^2 \sigma, O\left(\frac{k\alpha}{\gamma}\right) \right)_k$-balanced partition of $D$. Moreover, $\mathcal{B}$ can be computed in $O(nm)$ time.

**Overview of Algorithms 1 and 2.** Algorithm 2 gets as input an $n \times m$-signal $D$ and two parameters $\sigma, \gamma$. Algorithm 2 aims to compute a balanced partition of $D$; see Fig. 2. In turn, it calls Algorithm 1, which takes as input an $n \times m$ sub-signal $R$ that is defined by several contiguous rows of the original dataset $D$, and a parameter $\sigma > 0$, and aims to compute a partition $\mathcal{B}$ of $R$. To do so, Algorithm 1 partitions $R$ along the vertical dimension (e.g., into vertical slices), in a greedy fashion, such that for every $B \in \mathcal{B}$, $\text{opt}_1(B)$ is as large as possible, while still upper bounded by $\sigma$. This will ensure that the partition is into a relatively small number of slices. In the case where one of the sub-signals $B$ in this vertical partition of $R$ contains only one column, and already exceeds the maximum tolerance $\text{opt}_1(B) > \sigma$, we recursively apply Algorithm 1 to $B^T$ in order to partition $B$ horizontally. As long as the total number of slices returned by Algorithm 1 is smaller than $1/\gamma$, Algorithm 2 adds yet another row to the previous set of rows, and repeats the above process. At this point, the partition of the current horizontal slice (collection of rows) $R$ is final, and is added to the output partition of Algorithm 2. In turn, a new horizontal slice $R$ of just one row, the first row of $D$ that is not included in the previous $R$, is initiated on which we again call Algorithm 1.

---

**Algorithm 1:** $\text{SLICEPARTITION}(D, \sigma)$

**Input** : A parameter $\sigma > 0$ and an $n \times m$ signal $D = \{(x_i, y_i)\}_{i=1}^{N}$.

**Output**: A partition $\mathcal{B}$ of $D$.

1  $\mathcal{B} := \emptyset$ and $c_{begin} := 1$
2  **while** $c_{begin} \leq m$ **do**
3     $B := \{((i,j), y) \in D \mid j = c_{begin}\}$
    // extract first column
4     **if** $\text{opt}_1(B) > \sigma$ **then**
5        $\mathcal{B}' := \text{SLICEPARTITION}(B^T, \sigma)$
6        $\mathcal{B} := \mathcal{B} \cup \left\{ B'^T \mid B' \in \mathcal{B}' \right\}$
      $c_{begin} := c_{begin} + 1$
7     **else**
8        $c_{end} := c_{begin}$
9        **while** $\text{opt}_1(B) \leq \sigma$ *and* $c_{end} < m$ **do**
10          $c_{end} := c_{end} + 1$ and $lastB := B$
11          $B := \{((i,j), y) \in D \mid i \in [c_{begin}, c_{end}]\}$
12       $\mathcal{B} := \mathcal{B} \cup \{lastB\}$
13       $c_{begin} := c_{end}$
14 **return** $\mathcal{B}$

---

**Algorithm 2:** $\text{PARTITION}(D, \gamma, \sigma)$

**Input** : An $n \times m$ signal $D$, a parameter $\gamma \in (0, 1)$, and a lower bound $\sigma \in [0, \text{opt}_k(D)]$.

**Output** : A partition $\mathcal{B}$ of $D$; see Lemma 7

1  $\mathcal{B} := \emptyset$ and $r_{begin} := 1$
2  **while** $r_{begin} \leq n$ **do**
3     $R := \{((i,j), y) \in D \mid i = r_{begin}\}$
    // extract first row
4     $\mathcal{B}' := \text{SLICEPARTITION}(R, \gamma^2 \sigma)$
5     $r_{end} := r_{begin}$
6     $last\mathcal{B}' := \mathcal{B}'$
7     **while** $|\mathcal{B}'| \leq 1/\gamma$ *and* $r_{end} < n$ **do**
8        $r_{end} := r_{end} + 1$
9        $last\mathcal{B}' := \mathcal{B}'$
10       $S := \{((i,j), y) \in D \mid i \in [r_{begin}, r_{end}]\}$
      // extract a slice
11       $\mathcal{B}' := \text{SLICEPARTITION}(S, \gamma^2 \sigma)$
12    $\mathcal{B} := \mathcal{B} \cup last\mathcal{B}'$
13    $r_{begin} := r_{end}$
14 **return** $\mathcal{B}$

# 4   Coreset Construction

In this section, we present our main algorithm (Algorithm 3), which outputs a $(k, \varepsilon)$-coreset for a given $n \times m$ signal $D$, the number of leaves $k \geq 1$, and an approximation error $\varepsilon \in (0, 1)$.

**Overview of Algorithm 3:** The algorithm first utilizes the $(\alpha, \beta)_k$-approximation from Section 2 to obtain a lower bound $\sigma \leq \mathrm{opt}_k(D)$ for the optimal $k$-segmentation. It then computes, as described in Section 3, a balanced partition $\mathcal{B}$ of $D$, where $\mathrm{opt}_1(B)$ is small and depends on $\sigma$, for every $B \in \mathcal{B}$. Finally, it computes a small representation $(C_B, u_B)$ for every $B \in \mathcal{B}$, and returns the union of those representations. Each such pair $(C_B, u_B)$ satisfies: (i) $|C_B| = 4$, and (ii) has the same weighted sum of values, weighted sum of squared values, and sum of weights, as $B$, i.e., $\sum_{(a,b) \in C_B} u_B((a, b)) \cdot (b \mid b^2 \mid 1) = \sum_{(x,y) \in B} (y \mid y^2 \mid 1)$; see Fig. 3. Such a representation can be computed using Caratheodory's theorem, as explained in Section E of the supplementary material.

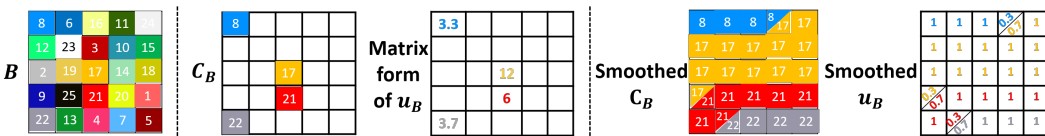

Figure 3: **(Left):** A matrix representing of a $5 \times 5$ sub-signal $B$ where $y$ is mapped into unique colors for every $(x, y) \in B$. **(Middle):** A representative (coreset) pair $(C_B, u_B)$ for $B$ where $C_B \subseteq B$ is a (small) subset and $u_B : C_B \rightarrow [0, \infty)$ is a weight function. That is, the pair $(C_B, u_B)$ satisfies $\sum_{(a,b) \in C_B} u_B((a, b)) \cdot (b \mid b^2 \mid 1) = \sum_{(x,y) \in B} (y \mid y^2 \mid 1)$. **(Right):** A duplication of the coreset points according to their weight. We call the resulting pair a "smoothed" version of $(C_B, u_B)$; see more details and formal definition in Section D of the supplementary material.

**Some intuition behind Algorithm 3:** Consider some $k$-segmentation $s$. By the properties of the balanced partition $\mathcal{B}$ of $D$, only a small number of sub-signals $B \in \mathcal{B}$ are intersected by $s$, i.e., assigned at least 2 distinct values. For every non-intersected sub-signal $B \in \mathcal{B}$, the loss $\ell(B, s)$ is accurately estimated by the (coreset) pair $(C_B, u_B)$. On the other hand, for every sub-signal $B \in \mathcal{B}$ which is intersected by $s$, by the guarantees of the representation $(C_B, u_B)$, the loss $\ell(B, s)$ will be approximated, using only $(C_B, u_B)$, up to some small error that depends on $\mathrm{opt}_1(B)$. However, again by the properties of $\mathcal{B}$, we have that $\mathrm{opt}_1(B)$ is small. Hence, using the union $(C, u)$ of the representations we can approximate $\ell(D, s)$ as required. Furthermore, combining that $|C_B| \in O(1)$ for every $B \in \mathcal{B}$ with the fact that $|\mathcal{B}|$ is small yields that $|C|$ is indeed small; see Theorem 8.

---

**Algorithm 3:** SIGNAL-CORESET$(D, k, \varepsilon)$; see Theorem 8

---

**Input** : An $n \times m$ signal $D$, an integer $k \geq 1$, and an error parameter $\varepsilon \in (0, 1/4)$.
**Output** : A $(k, \varepsilon)$-coreset $(C, u)$ for $D$.

1   $s :=$ an $(\alpha, \beta)_k$ approximation of $D$ for $\alpha \in O(k \log(nm))$ and $\beta \in k^{O(1)} \log^2(nm)$ ; see Lemma 5 for suggested implementation.
2   $\gamma := \varepsilon^2 / (\beta k)$, $\sigma := \frac{\ell(D,s)}{\alpha}$ and $C := \emptyset$
3   $\mathcal{B} :=$ PARTITION$(D, \gamma, \sigma)$ // see Algorithm 2.
4   **for** *every set $B \in \mathcal{B}$* **do**
5     $(C_B, u_B) :=$ a $(1, 0)$-coreset for $B$, (a zero error coreset for $k = 1$), such that $C_B \subseteq B$, $|C_B| = 4$, and $\sum_{(a,b) \in C_B} u_B((a, b)) \cdot (b \mid b^2 \mid 1) = \sum_{(x,y) \in B} (y \mid y^2 \mid 1)$ this is done using Caratheodory's theorem; see Corollary 17 in the appendix.
6     Replace each of the coordinates $a$ of the 4 pairs $(a, b) \in C$ with one of the 4 corner coordinates of the pairs in $B$ ; see detailed explanation if the proof of Theorem 8.
7     $C := C \cup C_B$ and $u((a, b)) := u_B((a, b))$ for every $(a, b) \in C_B$.
8   **return** $(C, u)$

---

**Theorem 8** (Coreset). *Let $D = \{(x_1, y_1), \cdots, (x_N, y_N)\}$ be an $n \times m$ signal i.e., $N := nm$. Let $k \geq 1$ be an integer (that corresponds to the number of leaves/rectangles), and $\varepsilon \in (0, 1/4)$ be an error parameter. Let $(C, u)$ be the output of a call to* SIGNAL-CORESET$(D, k, \varepsilon/\Delta)$ *for a*

*sufficiently large constant* $\Delta \geq 1$*; see Algorithm 3. Then,* $(C, u)$ *is a* $(k, \varepsilon)$*-coreset for D of size* $|C| \in \frac{(k \log(N))^{O(1)}}{\varepsilon^4}$*; see Definition 3. Moreover,* $(C, u)$ *can be computed in* $O(kN)$ *time.*

**Coreset size.** While Theorem 8 gives a worst-case theoretical upper bound, this bound is too pessimistic in practice, as common in coreset papers [42, 34]. This phenomenon is well known for coresets; see discussion e.g., in [21, 53]. The reasons might include: worst-case artificial examples vs. average behaviour on structured real-world data, noise removing/smoothing by coresets, the fact that in practice we run heuristics that output a local minima (and not optimal solutions with global minimum), non-tight analysis (especially when it comes to constants), etc.

Our experiments in Section 5 show that, empirically, the constructed coresets are significantly smaller: for $N := nm \sim 140,000$, $k = 1000$, and $\varepsilon = 0.2$, Theorem 8 predicts, in the worst case, a coreset of size larger than the full dataset size $N$. However, such an $\varepsilon$ error is obtained with a coreset of size at most 1% of the input; see Fig. 4.

## 5    Experimental Results

We implemented our coreset construction from Algorithm 3 in Python 3.7, and in this section we evaluate its empirical results, both on synthetic and real-world datasets. More results are placed in the supplementary material; see Section A. Open-source code can be found in [35]. The hardware used was a standard MSI Prestige 14 laptop with an Intel Core i7-10710U and 16GB of RAM. Since our coreset construction algorithm does not compete with existing solvers, but improves them by reducing their input as a pre-processing step, we apply existing solvers as a black box on the small coreset returned by Algorithm 3. The results show that our coreset can boost, by up to x10 times, the running time and storage cost of common random forest implementations.

**Implementations for forests.** We used the following common implementations: (i) the function `RandomForestRegressor` from the `sklearn.ensemble` package, and (ii) the function `LGBMRegressor` from the `lightGBM` package that implements a forest of gradient boosted trees. Both functions were used with their default hyperparameters, unless states otherwise.

**Data summarizations.** We consider the following compression schemes:

**(i):** `DT-coreset`$(D, k, \varepsilon)$ - The implementation based on Algorithm 3. In all experiments we used a constant $k = 2000$ for computing the coreset, regardless of the (larger) actual $k$ value in each test, since $k = 2000$ was sufficient to obtain a sufficiently small empirical approximation error. Hence, the parameter $\varepsilon$ controls the trade-off between size and accuracy.

**(ii):** `RandomSample`$(D, \tau)$ - returns a uniform random sample of size $\tau$ from $D$. In all tests $\tau$ was set to the size of the coreset `DT-coreset`$(D, k, \varepsilon)$ for fair comparison.

**Datasets.** We used the following pair of datasets from the public UCI Machine Learning Repository [3], each of which was normalized to have zero mean and unit variance for every feature:

**(i): Air Quality Dataset** [18] - contains $n = 9358$ instances and $m = 15$ features.

**(ii) Gesture Phase Segmentation Dataset** [45] - contains $n = 9900$ instances and $m = 18$ features.

**The experiment.** The goal was to predict missing entries in every given dataset, by training random forests on the available data. The test set (missing values) consists of 30% of the dataset, and was extracted from the input dataset matrix by randomly and uniformly choosing a sufficient number of $5 \times 5$ patches in the input dataset, and defining them as missing values. The final loss of a trained forest is the sum of squared distances between the forest predictions for the missing values, and the ground truth values. To tune the hyperparameter $k$, we randomly generate a set $\mathcal{K}$ of possible values for $k$ on a logarithmic scale. Then, we either: (i) apply the standard tuning (train the forest on the full data, for each value in $\mathcal{K}$, and pick the one with the smallest test set error), or (ii) compress the input (only once) into a small representative set, and then apply the standard tuning on the small, rather than the full, data. The experiment was repeated 10 times. All the results are averaged over all 10 tests; see Fig. 4.

**Discussion.** While the size and accuracy of our coreset are independent of our exact implementation of Algorithm 3, the running time is heavily based on our naive implementation, as compared to the very efficient professional Python libraries. This explains why most of the running time is still devoted to the coreset construction rather than the forest training. Nevertheless, even our simple implementation yielded improvements of up to x10 in both computational time and storage, for a relatively small accuracy drop of 0.03 in the SSE. Tuning more than one hyperparameter will result in a bigger improvement. Furthermore, Fig. 4 empirically shows that tuning a hyperparameter on the

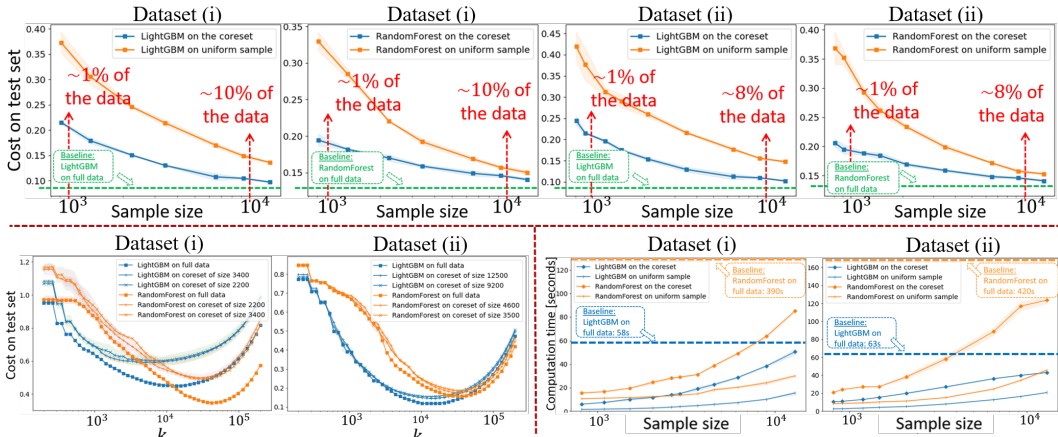

Figure 4: **Experimental results. (Top):** The $X$-axis is the compression size. For every compression size $\gamma$, hyperparameter tuning is applied on both the coreset and the uniform sample (which are both of size $\gamma$). A random forest is then trained, on the full data, using those tuned parameters. The $Y$-axis presents the test set SSE loss of the trained forests. **(Bottom left):** Hyperparameter tuning. For every different value of $k$ ($X$-axis), a forest is trained using this parameter value either on the compression (of two different sizes) or on the full data. The $Y$-axis presents $\ell + k/10^5$, where $\ell$ is the normal SSE loss of the trained forest on the test set. **(Bottom right):** Time comparison. The $Y$-axis presents the total running time of both to compute the compression and to tune the parameter $k$ on the compression (out of 50 different values). Note that the bottom right figures measure the total time to tune the parameter $k$ in the bottom left figures, but using many more compression sizes. The optimal obtained parameter was then used to train the random forest in the top figures.

coreset yields a loss curve very similar to the loss curve of tuning on the full data. Lastly, we observe that, in practice, our coresets have size much smaller than predicted in the pessimistic theory.

## 6   Conclusions and Future Work

While coresets for $k$-trees do not exist in general, we provided an algorithm that computes such a coreset for every input $n \times m$ *signal*. The coreset size depends polynomialy on $k \log(nm)/\varepsilon$ and can be computed in $O(nmk)$ time. Our experimental results on real and synthetic datasets demonstrates how to apply existing forest implementations and tune their hyperparameters on our coreset to boost their running time and storage cost by up to x10. In practice our coreset works very well also on non-signal datasets, probably since they have "real-world" properties that do not exist in the artificial worst-case example from Section 1.2. An open problem is to define these properties. Moreover, while this paper focuses on the sum of squares distances loss, we expect that the results can be generalized to support other loss functions; see Section 1.2. Lastly, supporting high-dimensional data (tensors), instead of matrices, is a straightforward generalization that can be achieved via minor modifications to our algorithms. We also leave this to future work.

## Acknowledgments and Disclosure of Funding

This research was supported by The ISRAEL SCIENCE FOUNDATION, grant number 379/21.

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
