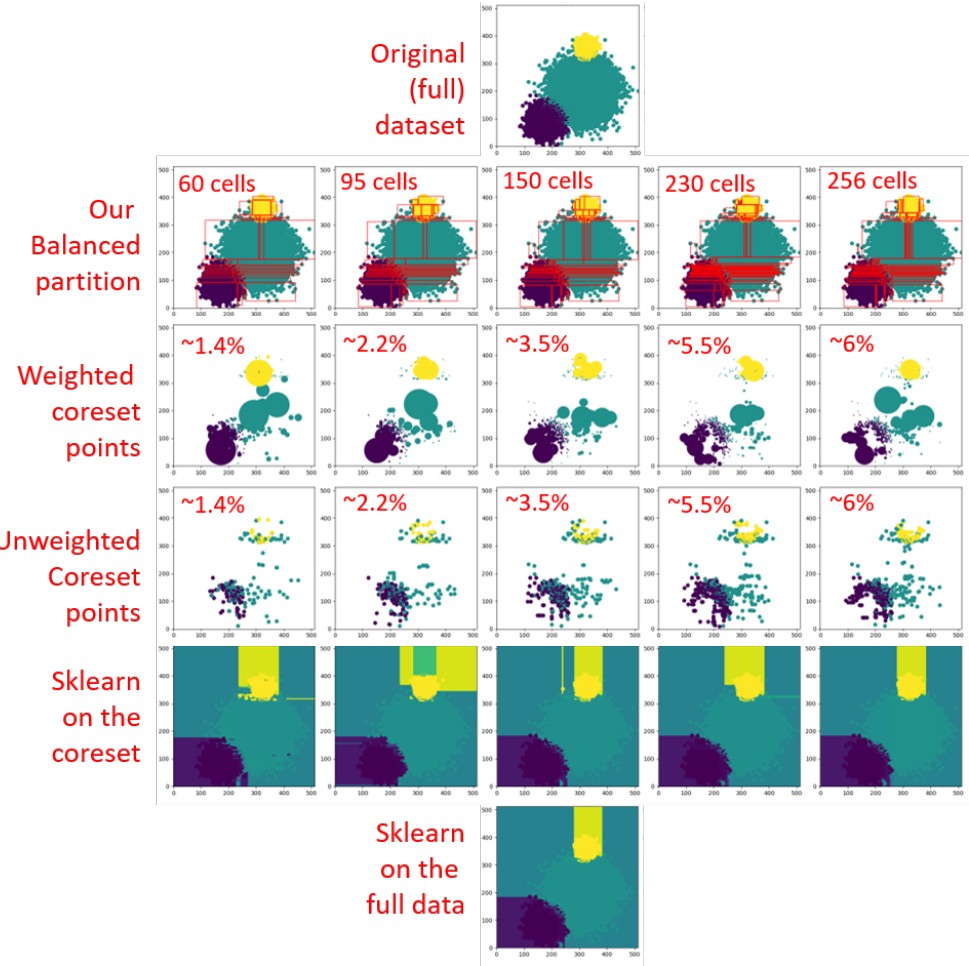

Figure 5: The blobs dataset. The dataset $D$ was generated using the function `sklearn.datasets.make_blobs`, and contains $n = 17,000$ points clustered into 3 clusters (containing $8500, 5800,$ and $2700$ points), each with a different label. A coreset $(C, u)$ was constructed using Algorithm 3. From top down, the rows illustrate: (i) The input dataset $D$, (ii) The balanced partition of $D$, including the number of sets in the partition (iii) The weighted coreset points. Each point $(x, y) \in C$ is plotted at location $x$, colored according to its label $y$, and its radius is proportional to its weight $u(x, y)$. The percentage of the coreset size relative to the full data is presented. (iv) The unweighted coreset points. Each point $(x, y) \in C$ is plotted at location $x$, colored according to $y$, and has a fixed radius. The percentage of the coreset size relative to the full data is presented. (v) The partition of the space via a decision tree computed using a call to `DecisionTreeRegressor` from the `sklearn.tree` package, where the input was the weighted coreset points only. Each region is assigned a color according to the label assigned to it by the computed tree. (vi) Similar to Row (v), but where the decision tree is trained on the full data. Algorithm 5 was used during training of the decision tree to evaluate the loss of each model.

## A   Additional Experiments

In this section, we present some additional experiments conducted using our algorithm from Sections 3-4. We give visual illustration both of our coreset itself and of the result of applying a very common decision tree implementation on the coreset, as compared to running the same function on the original (full) data; see Fig. 5-6.

**Discussion.   Visual representation.** As seen in Fig. 5-6, the balanced partition in the second row partitions the input data into multiple subsets, where, as expected, flat and relatively smooth regions are partitioned into a smaller number of large cells, while more complex regions are partitioned into a

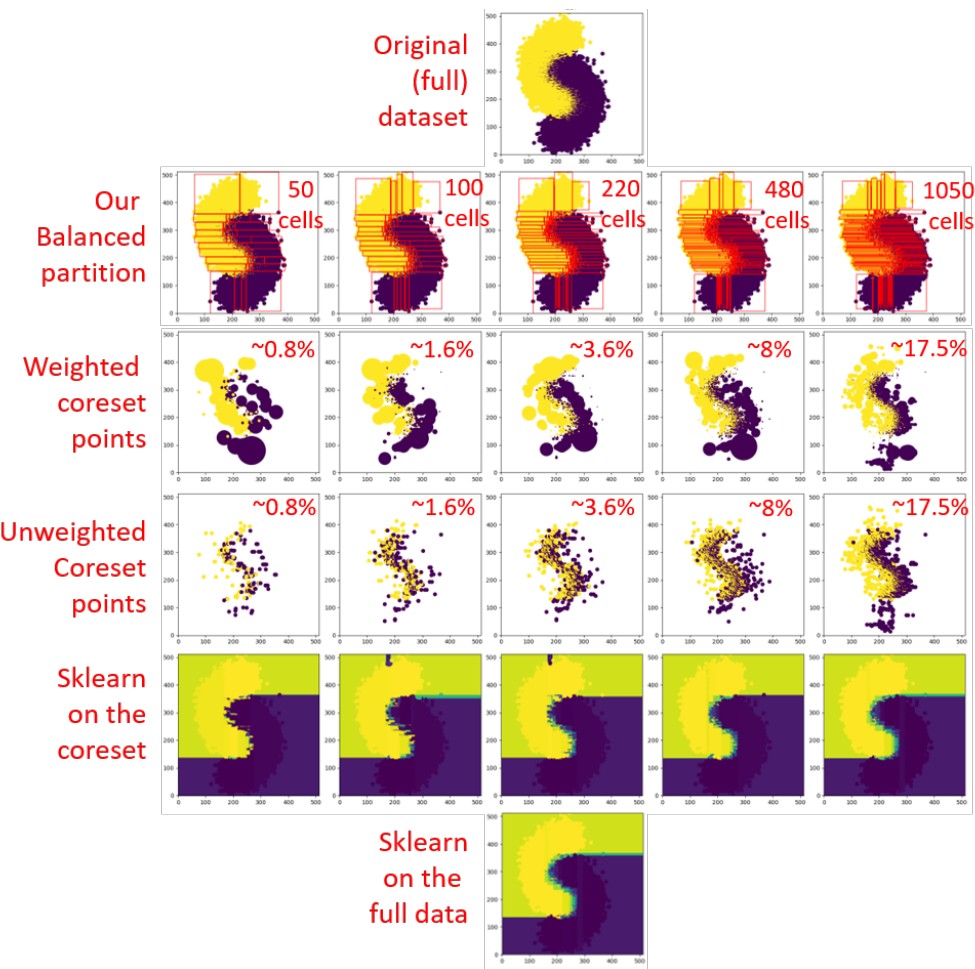

Figure 6: The moons dataset. The dataset was generated using the function `sklearn.datasets.make_moons`. The dataset contains $n = 24,000$ points spread across two interleaving half circles ($12,000$ points for each half circle), each with a different label. See caption of Fig. 5 for a detailed explanation about the rows.

larger number of finer cells. This is expected since the balanced partition insures a small variance inside each cell.

Furthermore, as seen in the third row of the above figures, the weighted coreset contains a small number of "large" circles (points with large weight) in the flat and relatively smooth regions, while it contains large number of "small" circles (points with small weight) in the more complex regions of the input.

**Accuracy.** As seen in the last two rows of Fig. 5-6, the decision tree trained only on the coreset points resembles the decision tree trained on the full data, even for coresets of size only $6\%$, $8\%$, and $14\%$ of the input data, as seen in Fig. 5,6, and 7 respectively. This implies a x10 faster training time of a decision tree (or, similarly, a forest) on a given coreset, compared to training it on the full data, with almost no compromises to the accuracy.

The difference in the coreset size required in order to accurately represent the full data depends on the complexity of the input dataset. Indeed, the dataset in Fig. 5 is a much simpler dataset for a decision tree to classify, compared to the dataset in Fig. 7. Hence, the coreset sizes required in Fig. 5 are smaller than the ones in Fig. 7.

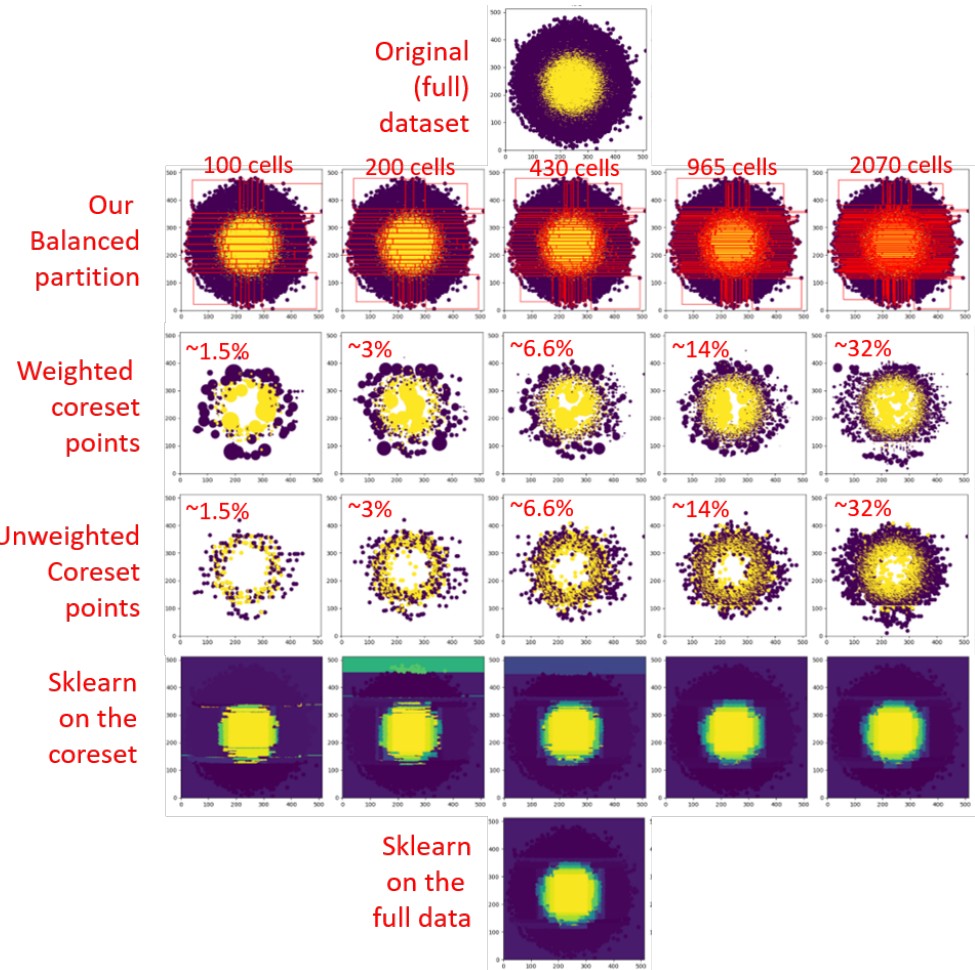

Figure 7: The circles dataset. The dataset was generated using the function `sklearn.datasets.make_circles`. The dataset contains $n = 26,000$ points spread across a big circle ($14,000$ points) and a small circle ($12,000$ points), each with a different label. See caption of Fig. 5 for a detailed explanation about the rows.

# B   Bi-criteria Approximation

**Notations.** A sub-signal $B$ is said to be *horizontally intersected* by a $k$-segmentation function $s$ if there are $((i_1, j_1), y_1), ((i_2, j_2), y_2) \in B$ where $i_1 \neq i_2$ such that $s(i_1, j_1) \neq s(i_2, j_2)$. Similarly, a block $B$ of $D$ is said to be *vertically intersected* by $s$ if there are $((i_1, j_1), y_1), ((i_2, j_2), y_2) \in B$ where $j_1 \neq j_2$ such that $s(i_1, j_1) \neq s(i_2, j_2)$. $B$ is said to be *intersected* by $s$ if $B$ is either horizontally or vertically intersected, i.e., $|\{s(x) \mid (x, y) \in B\}| > 1$. A set of sub-signals $\mathcal{B}$ is said to be horizontally (vertically) intersected by $s$ if it contains a sub-signal $B \in \mathcal{B}$ that is horizontally (vertically) intersected by $s$.

Also, we might abuse notation and denote by signal (sub-signal) an $n \times m$ signal (sub-signal) and by $k$-segmentation an $n \times m$ $k$-segmentation.

In this section we give a constructive proof for Lemma 5. A suggested implementation for this constructive proof is given in Algorithm 4.

We first prove a small technical observation (see Observation 9), and then we prove Lemma 10, which will be used throughout the proof of Lemma 5.

**Observation 9.** *Let $A$ and $B$ be two $n \times m$ sub-signals. Then it holds that*

$$\text{opt}_1(A \cup B) \geq \text{opt}_1(A) + \text{opt}_1(B).$$

**Algorithm 4:** BICRITERIA$(D, k)$; Lemma 5

---

**Input** : An $n \times m$ sub-signal $D = \{(x_i, y_i)\}_{i=1}^N$ and an integer $k \geq 1$.
**Output**: An $(\alpha, \beta)_k$-approximation for $D$.

1   $\mathcal{B} := \emptyset$
2   $\nu, \gamma :=$ sufficiently large constants // see proof of Lemma 5
3   **while** $|D| > k \log N$ **do**
4     **if** $D$ *contains a row* $R$ *with* $|R| \geq \frac{|D|}{\nu k}$ **then**
5       Partition $[m]$ into $t' = \gamma k$ intervals $[m] = \cup_{j=1}^{t'} I_j$ such that every $j \in [t']$, the size of
       each corresponding sub-signal $R_j = \{((x_1, x_2), y) \in R \mid x_2 \in I_j\}$ is
       $|R_j| \in \left\{ \frac{|R|}{t'} - 1, \frac{|R|}{t'} + 1 \right\}$. // e.g., by a greedy pass over $[m]$.
6       $\mathcal{B} :=$ the set of $t' - 2k$ signals $R_j$ with the smallest $\mathrm{opt}_1(R_j)$.
7     **else**
8       Partition $[n]$ into $\psi$ intervals $[n] = \cup_{j=1}^{\psi} I_j$ such that for every $j \in [\psi]$, the size of each
       corresponding sub-signal $D_j = \{((x_1, x_2), y) \in D \mid x_1 \in I_j\}$ is $\frac{|D|}{\nu k} \leq |D_j| \leq \frac{2|D|}{\nu k}$.
       // e.g., by a greedy pass over $[n]$.
9       **if** *at least* $\psi/2$ *of the sub-signals* $D_j$ *do not contain a column col of size* $|col| \geq \frac{|D_j|}{2(\nu k)^2}$
       **then**
10         Vertically partition each of the (at least $\psi/2$) sub-signals $D_j$ into $\psi_j$ sub-signals, each
         such sub-signal $B$ of size $\frac{|D_j|}{2(\nu k)^2} \leq |B| \leq \frac{|D_j|}{(\nu k)^2}$, and let $\mathcal{B}'$ contain the union of all
         those sub-signals. e.g., via a greedy algorithm.
11         $\mathcal{B} :=$ the set of $|\mathcal{B}'| - 4\nu^2 k^3 - 2k\psi$ signals $B \in |\mathcal{B}'|$ with the smallest $\mathrm{opt}_1(B)$.
12       **else**
13         $\mathcal{B} := \left\{ C \mid C \text{ is a column of } D_j, |C| \geq \frac{|D_j|}{2(\nu k)^2} \text{ and } j \in [\psi] \right\}$
14     $D := D \setminus \cup_{B \in \mathcal{B}'} B$ and $\mathcal{B} := \mathcal{B} \cup \mathcal{B}$
15   $s(b) := 1/|B| \sum_{(x,y) \in B} y$ for every $b \in B$ and $B \in \mathcal{B}$.
16   **return** $s$

---

*Proof.* Let $C = A \cup B$. Let $\mu = \frac{1}{|C|} \sum_{(x,y) \in C} y$ be the weighted mean of $A \cup B$. By Definition of opt we have that

$$
\begin{aligned}
\mathrm{opt}_1(A \cup B) &= \sum_{(x,y) \in C} (y - \mu)^2 \\
&= \sum_{(x,y) \in A} (y - \mu)^2 + \sum_{(x,y) \in B} (y - \mu)^2 \\
&\geq \mathrm{opt}_1(A) + \mathrm{opt}_1(B),
\end{aligned}
$$

where the first derivation holds since the mean of a points minimizes the sum of squared distances to those points, and the last derivation is by the definition of $\mathrm{opt}_1$.    $\square$

**Lemma 10.** *Let* $D = \{(x_1, y_1), \cdots, (x_N, y_N)\}$ *be an* $n \times m$ *sub-signal and let* $k \geq 1$ *be an integer. Then, in* $O(N)$ *time we can find a set* $\mathcal{B}$ *of* $|\mathcal{B}| = t \in O(k^3)$ *mutually disjoint blocks with respect to* $D$, *for which*

   *(i)* $\sum_{B \in \mathcal{B}} \mathrm{opt}_1(B) \leq \mathrm{opt}_k(D)$.

   *(ii)* $|\cup_{B \in \mathcal{B}} B| \in \Omega\left(\frac{N}{k}\right)$.

*Proof.* Let $\nu > 50$ be an arbitrary parameter and let $\gamma \geq 8$ be a parameter that will be defined later. We will prove Lemma 10 for $t \leq 2\nu^3 k^3$ and for $|\cup_{B \in \mathcal{B}} B| \geq \frac{N}{8\nu k}$.

We start with the simple 1-dimensional case, namely – we assume that $m = 1$. In this case, we just partition $[n]$ into $t' = \gamma k$ consecutive intervals $[n] = \cup_1^{t'} E_j$, such that each corresponding sub-signal $D_j = \{((a, b), y) \in D \mid a \in E_j\}$, $j \in [t']$ of $D$ has equal share of elements in $D$ (up to $\pm 1$), i.e.,

$|D_j| \in \{|D|/t' - 1, |D|, |D|/t' + 1\}$. This can be done by moving point by point along the elements of $D$, which are assumed to be sorted in ascending order according to $a$ for every $((a,b),y) \in D$, and defining a new interval at the first moment the current interval contains more than $|D|/t'$ elements of $D$ or just at the prior point. We then compute for each $D_j$, $j \in [t']$ the optimal $\mathrm{opt}_1(D_j)$, and return as output the set $\mathcal{B}$ containing the $t = t' - 2k$ sub-signals $D_j$ with the smallest loss $\mathrm{opt}_1(D_j)$, among all the $t'$ sub-signals $\{D_1, \cdots, D_{t'}\}$.

To see what guarantees we get, note that the computation takes $O(nm) = O(N)$ time to sort the elements $((a,b),y)$ of $D$ according to $a$, since $a \in [nm]$ is a bounded integer. Afterwards, the above greedy partition also takes $O(nm) = O(N)$ time. Furthermore,

$$|\cup_{B \in \mathcal{B}} B| \geq (\gamma - 2)k \cdot \frac{|D|}{\gamma k} \geq \frac{\gamma - 2}{\gamma}|D| \geq \frac{N}{8\nu k}.$$

For $\gamma \geq 8$ this proves Property (ii) above.

Finally, any $n \times 1$ $k$-segmentation function intersects at most $2k$ sub-signals from $\{D_j\}_{j=1}^{t'}$ (i.e., at most $2k$ sub-signals are assigned more than 1 distinct value via the $k$-segmentation function). Hence, the optimal $k$-segmentation $s^*$ of $D$ intersects at most $2k$ of those sub-signals as well. This implies that at least $t' - 2k \geq (\gamma - 2)k$ of the intervals $\{D_j\}_{j=1}^{t'}$ which are assigned 1 distinct value $|\{s(x) \mid (x,y) \in D_j\}| = 1$. Hence, by Observation 9, we conclude that $\mathrm{opt}_k(D) \geq \sum_{B \in \mathcal{B}} \mathrm{opt}_1(D)$ which verifies the 1st item above.

We remark that for the 1-dimensional case we can do much better (there is an overall $(1 + \varepsilon)$-approximation of the $k$-segmentation using logarithmic number of blocks), but this will not be used here. Another remark is that we have ignored the $\pm 1$ slack in the sizes above, making the actual part of $D$ that is removed at least $\frac{\gamma - 2}{\gamma}|D| - t'$. This is insignificant in the 1-dimensional case above, as for $t = O(1)$ and for $|D| \geq \log n$ this would be an insignificant fraction, while for smaller $D$, we can just use single point sub-signals.

**The 2-dim case** : Consider a row $R$ of $D$, say the $i'$th row $R = \{((a,b),y) \in D \mid a = i\}$. We call a row $R$ of $D$ $r$-heavy if $|R| \geq |D|/r$, namely – $R$ contains at least $|D|/r$ elements from $D$. Analogously, we define a column to be $r$-heavy.

Assume first that our $D$ contains a $\nu k$-heavy row $R$. We choose $R$ and use it as in the 1-dimensional case. As explained above we can find in $R$ a set of disjoint sub-signals $\mathcal{B}$ containing $\gamma k$ blocks and for which the first item above holds for $\mathrm{opt}_k(R)$ and in particular for $\mathrm{opt}_k(D)$, i.e., $\sum_{B \in \mathcal{B}} \mathrm{opt}_1(B) \leq \mathrm{opt}_k(R) \leq \mathrm{opt}_k(D)$. Further, using the above guarantees for the 1-dim case, $|\cup_{B \in \mathcal{B}} B| \geq \frac{\gamma - 2}{\gamma} \cdot |R| \geq \frac{\gamma - 2}{\gamma} \cdot \frac{|D|}{\nu k}$. This proves 2nd item for this case (with $\gamma \geq 3$).

Otherwise, let $e_i = |R_i|$ where $R_i = \{((a,b),y) \in D \mid a = i\}$. By our assumption $e_i \leq |D|/\nu k$ for every $i \in [n]$. Our algorithm is essentially identical to the 1-dimensional case, on the 1-dim array $L = (e_1, \ldots, e_{n_1})$ where we weight the $i$th element by $e_i$. Namely, we find a partition of $L$ into $\psi$ contiguous subintervals $\mathcal{E} = \{E_1, \cdots, E_\psi\}$ such that the corresponding sub-signals $D_j = \{((a,b),y) \in D \mid a \in E_j\}$ of $D$ are as equal as possible. By our assumption this could be done so that for any $j \in \psi$, the number of elements in $D_j$ is between $\frac{|D|}{\nu k}$ and at most $\frac{2|D|}{\nu k}$. This is since adding a new 'point' from the list to an existing interval may increase the sum by at most $|D|/\nu k$. this implies that $\nu k/2 \leq \psi \leq \nu k$.

Next we perform the above algorithm again on $D_1, \cdots, D_\psi$ with the intention to "vertically partition" each such $D_j$, i.e., split each $D_j$ into sets according to the value $b$ for every $((a,b),y) \in D_j$ (rather than considering the value $a$ above). Let $r = 2\nu^2 k^2$, we continue with the following case analysis: (i) at least a $\frac{1}{2}$-fraction of $\{D_1, \cdots, D_\psi\}$ contain no a $r$-heavy column, and (ii) at most a $\psi/2$ of $\{D_1, \cdots, D_\psi\}$ contain no a $r$-heavy column.

**Case (i):** At least a $\frac{1}{2}$-fraction of $\{D_1, \cdots, D_\psi\}$ contain no a $r$-heavy column. Then we partition each set $D_j$ with no $r$-heavy column into $\psi_i$ sub-signals $\left\{D_j^{(1)}, \cdots, D_j^{(\psi_i)}\right\}$ of nearly equal number of points, where the partition is applied onto the values $b$ of every $((a,b),y) \in D_j$. By a reasoning similar to that above, each $r/2 \leq \psi_i \leq r$, and each such block $B \in \left\{D_j^{(1)}, \cdots, D_j^{(\psi_i)}\right\}$ contains $|D_i|/r \leq |B| \leq 2|D_i|/r$. Using the bounds on $|D_i|$ and $r$ we get $\frac{|D|}{2\nu^3 k^3} \leq |B| \leq \frac{2|D|}{\nu^3 k^3}$. In particular

we conclude that the total number of such sub-signals is at most $t'$, and $t' \leq 2\nu^3 k^3$. On the other hand, $t' \geq \frac{\psi}{2} \frac{|D|}{\nu k} \cdot \frac{\nu^3 k^3}{2|D|} \geq \nu^3 k^3/8$. Let $\mathcal{B}_2$ be the collection of these sub-signals.

We choose for our output collection, $\mathcal{B}$, the set of $t' - z$ sub-signals $B \in \mathcal{B}_2$ with the smallest $\mathrm{opt}_1(B)$, for $z = 2k(r + \psi) \leq 6k^3\nu^2$.

We note that $\mathcal{B}$ contains $t$ sub-signals, where $t' - z \leq t \leq t'$. Further, by the lower bound on $t$ it follows that $|\cup_{B \in \mathcal{B}} B| \geq (t' - z) \cdot \frac{|D|}{2\nu^3 k^3} \geq (\nu^3 k^3/8 - 6k^3\nu^2)\frac{|D|}{2\nu^3 k^3} \geq \frac{|D|}{8}(1 - 48/\nu)$.

This verifies the 2nd item for this case. Finally, we note that any row of $D$ is shared by at most $r$ sub-signals of $\mathcal{B}$, and each column of $D$ is shared by at most $\psi$ sub-signals of $\mathcal{B}$. Hence, any $k$-segmentation function may intersect at most $z = 2k(r + \psi)$ sub-signals from $\mathcal{B}$. Therefore, for any $k$-segmentation $s$ there are at least $t' - z$ sub-signals in $\mathcal{B}_2$ which are not-intersected by $s$. By our definition of $\mathcal{B}$ to be the set of $t' - z$ sub-signals in $\mathcal{B}_2$ with the smallest loss, we obtain that the loss $\ell(D, s)$ is at least $\sum_{B \in \mathcal{B}} \ell(B)$ proving the 1st item of the lemma for this case.

**Case (ii):** At most $\psi/2$ of $\{D_1, \cdots, D_\psi\}$ contain no a $r$-heavy column, namely – at least $\psi/2$ of the $D_i$ have a $r$-heavy column. In this case we take the heavy column from each $D_i$ as its own sub-signal. We get a collection $\mathcal{B}_1$ of $\psi_1 \geq \psi/2$ blocks. We now return as output the set $\mathcal{B}$ of the $\psi_1 - 2k$ sub-signals $B \in \mathcal{B}_1$ with the smallest $\mathrm{opt}_1(B)$. We note that number of blocks we output in this case is at most $\psi \leq \nu k$.

Note also that $\mathcal{B}$ contains at least $\psi/2$ blocks, it follows that $|\cup_{B \in \mathcal{B}} B|) \geq \frac{\psi}{2} \cdot \frac{|D|}{2\nu^2 k^2} \geq \frac{|D|}{8\nu k}$ which proves the 2nd item in the lemma.

Finally, note that any $k$-segmentation $s$ can intersect at most $2k$ intervals from $\mathcal{B}_1$ (similarly to the 1-dim case). Hence, there are at least $|\mathcal{B}_1| - 2k = \psi_1 - 2k$ sub-signals in $\mathcal{B}_1$ that are not-intersected by $s$, which implies that its loss $\ell(D, s)$ is at least the sum $\sum_{B \in \mathcal{B}} \mathrm{opt}_1(B)$, which proves the 1st item of this lemma.

**Remark:** we did not optimise the parameter. A slightly better partition can be obtains (less blocks), but this is good enough for our purposes.

**Computational time:** Note that the elements of the input $D$ can be sorted in lexicographic order in $O(nm) = O(N)$ time since the coordinates $a$ and $b$ for every $((a, b), y) \in D$ are bounded integers. Then, a linear-time preprocessing can be applied to the input $D$ to store some statistics, e.g., the number of elements in each non-empty row or column, and the index of the next non-empty row or column for every element in $D$. Afterwards, the above greedy partition also takes $O(nm) = O(N)$ time. $\qquad \square$

We now restate and prove Lemma 5 from Section 2.

**Lemma 11** (Lemma 5). *Let $D = \{(x_1, y_1), \cdots, (x_N, y_N)\}$ be an $n \times m$ signal and $k \geq 1$ be an integer. Then, in $O(kN)$ time we can compute an $(\alpha, \beta)_k$-approximation for $D$, where $\alpha \in O(k \log N)$ and $\beta \in O(k^{O(1)} \log^2 N)$.*

The proof of the above claim is a constructive proof. A suggested implementation is provided in Algorithm 4.

*Proof.* The top level idea of the algorithm is as follows. We suggest an iterative algorithm. We start the first iteration with $D_1 = D$ and, using Lemma 10, find a collection of disjoint sub-signals $\mathcal{B}_1 = \{B_1, \cdots, B_t\}$ in $D_1$ (which will not necessarily cover the entire signal $D_1$), such that: (i) the sum of their 1-segmentation loss satisfies, $\sum_{i=1}^{t} \mathrm{opt}_1(B_i) \leq \mathrm{opt}_k(D_1) = \mathrm{opt}_k(D)$, and (ii) $\cup_{B \in \mathcal{B}_1} B$ has size $|\cup_{B \in \mathcal{B}_1} B| \geq |D_1|/c$ for some $c$ (e.g., $c \in O(k)$ in the lemma). Let $\mathcal{B}_1$ be such a collection. We then 'delete' from $D_1$ the elements of $\cup_{B \in \mathcal{B}_1} B$, and set $D_2 = D_1 \setminus \cup_{B \in \mathcal{B}_1} B$.

In the $i$th iteration, we repeat the same process with respect to the current set $D_i$. Namely, we find a collection $\mathcal{B}_i$ of at most $t$ disjoint sub-signals in $D$, for which: (i) $\sum_{B \in \mathcal{B}_i} \mathrm{opt}_1(B) \leq \mathrm{opt}_k(D_i) \leq \mathrm{opt}_k(D)$, and (ii) $\cup_{B \in \mathcal{B}_i} B$ has size $|\cup_{B \in \mathcal{B}_i} B| \geq |D_i|/c$, i.e., those blocks cover at least a constant fraction of $D_i$.

Repeating these iterative procedure for at most $\psi = O(c \log(nm))$ times, we end up covering all entries of $D$ with sub-signals where the overall loss of the sub-signals in each iteration is at most

$\mathrm{opt}_k(D)$. This defines a collection of at most $t\psi$ sub-signals $\mathcal{B}$ that cover the entire original set $D$. Hence, the total overall loss over those sub-signals is $\sum_{B \in \mathcal{B}'} OPT_1(B) \leq \psi \mathrm{opt}_k(D)$. By defining the output function $s$ such that for every $B \in \mathcal{B}$ and $b \in B$, $s$ assigns to $b$ the mean value of $B$, i.e., $s(b) = 1/|B| \sum_{((i,j),y) \in B} y$, we obtain that $\ell(B, s) = \mathrm{opt}_1(B)$ for every $B \in \mathcal{B}$, and that the $\ell(D, s) \leq \sum_{B \in \mathcal{B}} \ell(B, s) \leq \psi \mathrm{opt}_k(D)$.

While for every $B \in \mathcal{B}$ $s$ assigns the same value for every element $b \in B$, there is not necessarily a partition of $n \times m$ into $|\mathcal{B}| \in O(t\psi)$ distinct axis-parallel blocks that correspond to the sub-signals of $\mathcal{B}$. Therefore, $s$ is not necessarily a $|\mathcal{B}|$-segmentation function. However, looking at all possible intersections of the sub-signals in $\mathcal{B}$, it is easy to realize that the $t\psi$ sub-signals in $\mathcal{B}$ define a partition of $D$ into at most $O(t^2\psi^2)$ sub-signals $\mathcal{B}'$ that indeed correspond to a distinct partition of $n \times m$ into $|\mathcal{B}'|$ distinct axis-parallel blocks. Hence, $s$ is guaranteed to be a $|\mathcal{B}'|$-segmentation function.

The parameters that are guaranteed by the lemma are $c \in O(k)$, $t \in O(k^3)$. This implies that $\beta = |\mathcal{B}'| \in O(t^2\psi^2) = O(k^8 \log^2 nm)$ and $\alpha = \psi \in O(k \log nm)$.

**Computational time:** By Lemma 10, each of the $\psi$ iterations above takes time linear in the input size. The input size in the $i$'th iteration is $O(N((k-1)/k)^i)$ since at each iteration we remove at least a $1/k$ fraction of the input. Hence, the total running time is the sum of the geometric series $N \cdot \sum_{i \in \psi} ((k-1)/k)^i \in O(kN)$. $\qquad \square$

## C  Balanced Partition

In this section we give our full proof for Lemma 7. We first prove the following lemma regarding the output of Algorithm 1.

**Lemma 12.** *Let $D$ be an $n \times m$ sub-signal, and $\sigma > 0$ be a parameter. Let $\mathcal{B} = \{B_1, \cdots, B_{|\mathcal{B}|}\}$ be the output of a call to $\mathrm{SLICEPARTITION}(D, \sigma)$, where the sub-signals in $\mathcal{B}$ are numbered according to the order in which each of them was added to $\mathcal{B}$; see Algorithm 1. Then the following properties hold:*

*(i) $\mathcal{B}$ is a partition of $D$.*

*(ii) $\mathrm{opt}_1(B) \leq \sigma$ for every sub-signal $B \in \mathcal{B}$.*

*(iii) If $|\mathcal{B}| > 8k$ then for any $k$-segmentation $s$ that does not horizontally intersect $D$ we have that $\ell(D, s) \geq \left(\frac{|\mathcal{B}|}{4} - 2k\right) \sigma$.*

*(iv) $\mathcal{B}$ can be computed in $O(|D|)$ time.*

*Proof.* We consider the variables defined in Algorithm 1.

**Proof of (i):** By construction it immediately follows that $\mathcal{B}$ is a partition of $D$.

**Proof of (ii):** Consider a sub-signal $B \in \mathcal{B}$. We prove (ii) for each of the following cases: **Case (a):** $B$ was added to $\mathcal{B}$ at Line 12, and **Case (b):** $B$ was either added to $\mathcal{B}$ at Line 6

**Case (a):** In this case, by the condition at Line 9, $B$ must satisfy that $\mathrm{opt}_1(B) \leq \sigma$.

**Case (b):** In this case, $B$ was returned via a recursive call. Hence, this case holds trivially by Case (a) above.

Therefore, (ii) above holds by combining Cases (a)–(b).

**Proof of (iii):** Let $t = |\mathcal{B}|$ and assume for simplicity that $t$ is an even number. Recall that the index of each sub-signal in $\mathcal{B}$ indicates its order of insertion to $\mathcal{B}$, i.e., $B_1$ is the first sub-signal that was inserted to $\mathcal{B}$ and $B_t$ was the last such sub-signal to be inserted to $\mathcal{B}$.

Observe that each recursive call $\mathcal{B}' := \mathrm{SLICEPARTITION}(B^T, \sigma)$ at Line 5 returns at least $|\mathcal{B}'| \geq 2$ sub-signals. This is because the recursive call happens only when $\mathrm{opt}_1(B^T) = \mathrm{opt}_1(B) > \sigma$, which can only happen if $|\{(i,j) \mid ((i,j),y) \in B\}| > 1$, i.e., $B^T$ exceeds the maximum tolerance, and can indeed be partitioned into sub-signals. Hence, there are at least $t/4$ distinct pairs of consecutive sub-signals $B_i$ and $B_{i+1}$ that were both either computed via the recursive call or both were not computed via the recursive call. We now show that each such pair satisfies $\mathrm{opt}_1(B_i \cup B_{i+1}) > \sigma$.

Consider a pair of consecutive sub-signals $B_i$ and $B_{i+1}$ that were both not computed via the recursive call at Line 5. Let $B' \subseteq B_{i+1}$ contain the elements $((i,j),y) \in B_{i+1}$ with the smallest value of $i$ over all elements of $B_{i+1}$. By the greedy partition loop at Line 9 we obtain that $\mathrm{opt}_1(B_i \cup B') > \sigma$. We now have that

$$\mathrm{opt}_1(B_i \cup B_{i+1}) \geq \mathrm{opt}_1(B_i \cup B') > \sigma,$$

where the first inequality is by Claim 9.

Consider a pair of consecutive sub-signals $B_i$ and $B_{i+1}$ that were both computed via the recursive call at Line 5. Then, similarly to the previous argument, we obtain that $\mathrm{opt}_1(B_i \cup B_{i+1}) > \sigma$.

Now, let $s$ be a $k$-segmentation that does not horizontally intersect $\mathcal{B}$, i.e., it does not horizontally intersect any $B \in \mathcal{B}$. By the definition of $s$, there might be at most $2k$ sub-signals in $\mathcal{B}$ which are vertically intersected by $s$. Hence, among the $t/4$ distinct consecutive pairs of sub-signals discussed above there are at least $t/4 - 2k$ such pairs that are not intersected by $s$.

Since $\mathcal{B}$ is a partition of $D$, we have that $\ell(D,s)$ is at least the sum of $\mathrm{opt}_1(B_i \cup B_{i+1}) \geq \sigma$, over the above $t/4 - 2k$ non-intersected pairs of sub-signals. Hence,

$$\ell(D,s) \geq \left(\frac{t}{4} - 2k\right)\sigma = \left(\frac{|\mathcal{B}|}{4} - 2k\right)\sigma.$$

**Proof of (iv):** The greedy Algorithm 1 can be implemented so that it computes only $O(|D|)$ operations. The most costly operation is the computation of $\mathrm{opt}_1(B)$ for some sub-signal $B$. We now argue that this can be computed in $O(1)$ time. Let $B$ be a sub-signal and let $\mu_B = 1/|B| \sum_{(x,y) \in B} y$ be its mean value. Observe that

$$
\begin{aligned}
\mathrm{opt}_1(B) &= \sum_{(x,y) \in B} (y - \mu_B)^2 \\
&= \sum_{(x,y) \in B} y^2 + |B| \cdot \mu_B - 2\mu_B \sum_{(x,y) \in B} y.
\end{aligned}
\tag{1}
$$

By precomputing and storing some statistics at each of the signal's elements, then the three terms on the right hand side of (1) can all be evaluated in $O(1)$ time for any sub-signal $B$. Hence, the total running time of Algorithm 1 is linear in the input size.

$\square$

We now restate and prove Lemma 7 from Section 3.

**Lemma 13.** *Let $D$ be an $n \times m$ signal, $k \geq 1$ be an integer, $\varepsilon \in (0, 1/4)$ be an error parameter, and $s : [n] \times [m] \to \mathbb{R}$ be an $(\alpha, \beta)_k$-approximation of $D$. Define $\sigma := \frac{\ell(D,s)}{\alpha}$ and $\gamma := \frac{\varepsilon^2}{\beta k}$. Then algorithm $\mathrm{PARTITION}(D, \gamma, \sigma)$ outputs a partition $\mathcal{B}$ of $D$ that is an $\left(O\left(\frac{\alpha}{\gamma^2}\right), \gamma^2\sigma, O\left(\frac{k\alpha}{\gamma}\right)\right)_k$-balanced partition in $O(|D|)$ time.*

*Proof.* To prove that $\mathcal{B}$ is a $\left(O\left(\frac{\alpha}{\gamma^2}\right), \gamma^2\sigma, O\left(\frac{k\alpha}{\gamma}\right)\right)_k$-balanced partition as in Definition 6, we need to prove the following properties:

(i) $\mathrm{opt}_1(B) \leq \gamma^2\sigma$ for every $B \in \mathcal{B}$.

(ii) $\mathcal{B}$ is a partition of $D$ whose size is $|\mathcal{B}| \in O\left(\frac{\alpha}{\gamma^2}\right)$.

(iii) For every $k$-segmentation $\hat{s}$ there are $O\left(\frac{k\alpha}{\gamma}\right)$ sub-signals $B \in \mathcal{B}$ for which $\hat{s}$ assigns at least 2 distinct values, i.e., $|\{\hat{s}(x) \mid (x,y) \in B\}| \geq 2$.

**Proof of (i):** Observe that the output set $\mathcal{B}$ contains the the union of multiple output sets $\mathcal{B}' := \mathrm{SLICEPARTITION}(\cdot, \gamma^2\sigma)$ computed via calls to Algorithm 1. By Property (ii) of Lemma 12, every sub-signal $B \in \mathcal{B}'$ in such output set $\mathcal{B}'$ satisfies that $\mathrm{opt}_1(B) \leq \gamma^2\sigma$. Hence, Property (i) of Lemma 7 immediately holds.

**Proof of (ii):** By the greedy construction it holds that $\mathcal{B}$ is a partition of $D$. We now prove that the number of times that Line 12 was executed is $t \in O(\alpha/\gamma)$, i.e., the number of times we append a set of signals $last\mathcal{B}'$ to $\mathcal{B}$ is at most $O(\alpha/\gamma)$. Let $\mathcal{B}_1, \cdots, \mathcal{B}_t$ denote the set of sub-signals $last\mathcal{B}'$ in each of the $t$ executions of Line 12, i.e., $\mathcal{B}_1 := last\mathcal{B}'$ at the first time Line 12 was executed. Each such set is called a *horizontal set*.

Recall that $s \in \text{SEG}_{\beta k}$ is a $\beta k$-segmentation. First of all, by the definition of a $\beta k$-segmentation function there are at most $2\beta k$ sets among the horizontal sets $\mathcal{B}_1, \cdots, \mathcal{B}_t$ which can be horizontally intersected by $s$.

Consider two consecutive horizontal sets $\mathcal{B}_i, \mathcal{B}_{i+1}$ that are not horizontally intersected by $s$, and let $H = \bigcup_{B \in B_i \cup B_{i+1}} B$ be the union of all the sub-signals in $\mathcal{B}_i \cup \mathcal{B}_{i+1}$. We now argue that the loss $\ell(H, s)$ is at least $O(\gamma\sigma)$. Since $\mathcal{B}_i$ and $\mathcal{B}_{i+1}$ are two different horizontal sets, by the greedy construction we know that their union $H$ could have been partitioned via a call to $\mathcal{E} := \text{SLICEPARTITION}(H, \gamma^2\sigma)$ into a set $\mathcal{E} = \left\{ E_1, \cdots, E_{|\mathcal{E}|} \right\}$ of at least $|\mathcal{E}| \geq \frac{1}{\gamma}$ blocks. By substituting $D = H, k = \beta k, \mathcal{B} = \mathcal{E}$ and $\sigma = \gamma^2\sigma$ in Property (iii) of Lemma 12, for a $\beta k$-segmentation $s$, we have that

$$\ell(H, s) \geq \left( \frac{|\mathcal{E}|}{4} - 2\beta k \right) \gamma^2\sigma \geq \left( \frac{1}{4\gamma} - 2\beta k \right) \gamma^2\sigma$$

$$\geq \left( \frac{1}{4\gamma} - \frac{\beta k}{9\varepsilon^2} \right) \gamma^2\sigma = \left( \frac{1}{4\gamma} - \frac{1}{9\gamma} \right) \gamma^2\sigma$$

$$\geq \gamma\sigma/2,$$

where the second derivation holds for $\varepsilon \in (0, 1/3)$, and the third derivation is by the definition of $\gamma$.

Assume by contradiction that there are more than $\frac{2\alpha}{\gamma}$ such pairs of consecutive horizontal sets $\mathcal{B}_i, \mathcal{B}_{i+1}$, which are not horizontally intersected by $s$. The loss of those slices to $s$ would be bigger than $\frac{2\alpha}{\gamma} \cdot \frac{\gamma\sigma}{2} = \alpha\sigma = \ell(D, s)$, which is a contradiction. Therefore, the number of pairs of consecutive horizontal sets, which are not horizontally intersected by $s$, cannot exceed $O\left( \frac{\alpha}{\gamma} \right)$. Observe that the total number of horizontal sets that can be intersected by $s$ is at most $2\beta k$. Hence, the total number of horizontal sets is at most

$$m \in O\left( \frac{\alpha}{\gamma} + 2\beta k \right) \in O\left( \frac{\alpha}{\gamma} \right). \tag{2}$$

We now prove that the number of output cells is at most $|\mathcal{B}| \in O\left( \frac{\alpha}{\gamma^2} \right)$ in two steps. In step (i) we consider the horizontal sets that contain more than one row of $D$ and show that they contain a total of $O\left( \frac{\alpha}{\gamma^2} \right)$ sub-signals. In step (ii) we consider the horizontal sets that contain exactly one row of $D$ and prove that they also contain a total of $O\left( \frac{\alpha}{\gamma^2} \right)$ sub-signals.

**Step (i):** By (2), the total number of horizontal sets is at most $m \in O(\frac{\alpha}{\gamma})$. Therefore, the total number of horizontal slices that contain more than one row of $A$ is also at most $O\left( \frac{\alpha}{\gamma} \right)$. By the construction in Algorithm 2, each such horizontal set $\mathcal{B}_i$ with more than 1 row of $A$ is partitioned into at most $\frac{1}{\gamma}$ sub-signals. Hence, the total number of blocks in horizontal sets than contain more than one row of $D$ is at most $O\left( \frac{\alpha}{\gamma} \cdot \frac{2}{\gamma} \right) = O\left( \frac{\alpha}{\gamma^2} \right)$.

**Step (ii):** Consider all the horizontal sets $\mathcal{B}_i$ which contain one row of $D$, and which have been partitioned into $|\mathcal{B}_i| \leq 2\beta k \leq 1/\gamma$ blocks. The total number of blocks in such horizontal slices is thus bounded by the maximum number of horizontal slices $m \in O(\alpha/\gamma)$ times $1/\gamma$ for a total of at most $O(\alpha/\gamma^2)$ blocks.

For the rest of this step, we assume that all horizontal sets $\mathcal{B}_i$ have $|\mathcal{B}_i| \geq 2\beta k$. Let $G \subseteq [t]$ contain the indices of the horizontal sets which contain exactly one row of $D$, and let $i \in G$. Observe that $\mathcal{B}_i$ was computed, at some point, via a call $\mathcal{B}_i := \text{SLICEPARTITION}(\cup_{B \in \mathcal{B}_i} B, \gamma^2\sigma)$. Also, since the points $\cup_{B \in \mathcal{B}_i} B$ of $\mathcal{B}_i$ all have the same row index, observe that $\mathcal{B}_i$ cannot be horizontally intersected by $s$. Therefore, by substituting $D = \cup_{B \in \mathcal{B}_i} B, k = \beta k$ and $\sigma = \gamma^2\sigma$ in Property (iii) of Lemma 12

we obtain that

$$\ell(\mathcal{B}_i, s) \geq \left( \frac{|\mathcal{B}_i|}{4} - 2\beta k \right) \cdot \gamma^2 \sigma. \tag{3}$$

Furthermore, we have that

$$\alpha \cdot \sigma = \ell(D, s) \geq \sum_{i \in G} \ell(\mathcal{B}_i, s) \geq \sum_{i \in G} \left( \frac{|\mathcal{B}_i|}{4} - 2\beta k \right) \cdot \gamma^2 \sigma, \tag{4}$$

where the first derivation is by the definition of $\sigma$, the second derivation holds since $\{(x, y) \in B \mid B \in \mathcal{B}_i, i \in G\} \subseteq D$, and the third derivation is by (3). Rearranging terms in (4) concludes Step (ii) as

$$\sum_{i \in G} |\mathcal{B}_i| \leq \frac{4\alpha}{\gamma^2} + 8 \sum_{i \in G} \beta k \leq \frac{4\alpha}{\gamma^2} + 8t\beta k$$

$$\leq \frac{4\alpha}{\gamma^2} + \frac{8\alpha\beta k}{\gamma} \in O\left( \frac{\alpha}{\gamma^2} \right).$$

Therefore, the total number of blocks in horizontal sets that contain exactly one row of $D$ is at most $O\left( \frac{\alpha}{\gamma^2} \right)$.

**Proof of (iii):** By the properties above we have that: (i) there are at most $O(\alpha/\gamma)$ horizontal sets, and (ii) each horizontal set $\mathcal{B}_i$ either contains at most $O(1/\gamma)$ sub-signals, or all the points $((i, j), y) \in B$ of all the blocks $B \in \mathcal{B}_i$ have the same row index $i$. Let $\hat{s}$ be a $k$-segmentation. $\hat{s}$ can horizontally intersect all the (at most) $1/\gamma$ sub-signals of at most $k$ horizontal sets, and can vertically intersect at most 1 block from each of the $O(\alpha/\gamma)$ horizontal sets. Hence, the total number of intersected sub-signals is $O(k\alpha/\gamma)$.

**Computational time:** We now prove that $\mathcal{B}$ can be computed in $O(|D|)$ time. The computational time of Algorithm 2 is dominated by the computational time of Line 11 where we partition a slice $S$. Using Algorithm 1 we can partition each such slice $S$ in linear $O(|S|)$ time; see Lemma 12. Therefore, the naive implementation, i.e. by calling Algorithm 1 for every slice $S$, will result in $O(|D|^2)$ overall time, since many rows of $D$ participate many times in such a call to Algorithm 1.

However, we can implement Line 11 in $O(m)$ time, rather than $O(|S|)$ time, by preprocessing the input signal $D$, in linear time $O(|D|)$, and storing some statistics for every element $((i, j), y) \in D$. For example, one can store the sum of values and squared values over all elements $((i', j'), y')$ where $i' < i$ or $j' < j$. Using those values we can compute $\mathrm{opt}(B)$ in $O(1)$ time for every sub-signal $B$ of $D$. Now, using such statistics (and possibly more statistics), Line 11 can be implemented in $O(m)$ time via a greedy algorithm that iterates over the points of the last row $R = \{((i, j), y) \in D \mid i = r_{end}\}$ added to $S$ (i.e. with no need to iterate over other elements of $S$). We leave the small details to the reader.

$\square$

## D   Coreset Construction

In this section, we provide the proof of correctness for our main coreset construction algorithm presented in Algorithm 3; see Theorem 15. Furthermore, we provide an algorithm than gets as input a $k$-segmentation $s$, as well as a $(k, \varepsilon)$-coreset for some input dataset $D$, which was computed using Algorithm 3. The algorithm returns a $(1 + \varepsilon)$-approximation to the loss $\ell(D, s)$, in $O(k|C|)$ time; see Algorithm 5 and full details in Lemma 14.

In what follows, for an $n \times m$ sub-signal $B$ and a weight function $u : B \to [0, \infty)$, we abuse notation and denote $u((a, b))$ by simply $u(a, b)$ for $(a, b) \in B$.

**Some intuition behind Algorithm 5.** Given a $(k, \varepsilon)$-coreset $(C, u)$ for an input dataset $D = \{(x_1, y_1), \cdots, (x_N, y_N)\}$, and a $k$-segmentation $s$, the algorithm outputs a $(1 + \varepsilon)$-approximation to $\ell(D, s)$ in time that depends only on $k$ and $|C|$.

During the computation of $(C, u)$ in Algorithm 3, a partition $\mathcal{B}$ of $D$ was computed. Then, for every set $B$ in the partition $\mathcal{B}$, a representative pair $(C_B, u_B)$ for $B$ was computed and added to $C$.

To approximate the loss $\ell(D, s)$, we will approximate individually $\ell(B, s)$ for every $B \in \mathcal{B}$, and return the sum of those losses. Therefore, we now consider a single set $B \in \mathcal{B}$, and consider the following two cases.

**Case (i) :** $s$ assigns the same value for all the elements of $B$. Then, by construction, it is guaranteed that $\ell(B, s) = \sum_{(x,y) \in C_B} u_B(x, y)(s(y) - y)^2$. Therefore, in this case, $\ell(B, s)$ will be accurately estimated using $(C_B, u_B)$.

**Case (ii) :** $s$ assigns more than one unique value to the elements of $B$. In this case, if we ignore the computational time, we would ideally want to compute a "smoothed version" $(S, w)$ of $(C_B, u_B)$, as shown in Fig. 3 (see (9)- (11) below for formal details). Then, we would return the loss $\sum_{(x,y) \in S} w(x, y)(s(y) - y)^2$. However, computing $(S, w)$ is not necessary, since there are many subsets of $B$ in which all the elements $x \in B$ have simultaneously the same label in $S$ and are assigned the same value by $s$. Combining this with the fact that those subsets are of rectangular (simple) shape, we obtain that the loss over those subsets can be evaluated efficiently, as computed in Algorithm 5.

---

**Algorithm 5:** FITTING-LOSS$((C, u), s)$; see Lemma 14

**Input** : A $(k, \varepsilon)$-coreset $(C, u)$ which was returned from a call to
SIGNAL-CORESET$(D, k, \varepsilon/\Delta)$ in Algorithm 3, for some $n \times m$-signal $D$, $k \geq 1$,
$\varepsilon \in (0, 1)$ and a sufficiently large $\Delta \geq 1$.
A $k$-segmentation (or $k$-tree) $s$.
**Output :** A $(1 + \varepsilon)$-approximation to the loss $\ell(D, s)$.

1   $loss := 0$
2   **for** *every* 4 *consecutive elements* $\hat{C} = \{(a_i, b_i)\}_{i=1}^4$ *in C* **do**
3     Denote by $B$ the sub-signal that corresponds to $C'$. // By construction in
     Algorithm 3, the coordinates $a$ of the 4 elements $(a, b) \in \hat{C}$ are the
     corners of $B$.
4     $z := |\{s(x) \mid (x, y) \in B\}|$
5     **if** $z = 1$ // i.e., $s$ does not intersect $B$
6     **then**
7       $loss_{\hat{C}} := \sum_{(x,y) \in C_B} u_B(x, y)(s(x) - y)^2$. // note that $s(x_1) = s(x_2)$ for
       every $x_1, x_2 \in B$
8     **else**
     // In this case, $s$ intersects $B$
9      Denote by $S$ the partition that $s$ induces onto $[n] \times [m]$. // $S$ contains $|S| \leq k$
      subsets of $[n] \times [m]$.
10      $i := 1$
11      **for** *every* $S' \in S$ **do**
12       Denote by $\ell$ the label that $s$ assigns to the elements of $S'$ i.e., $s(x, y) = \ell$ for every
       $(x, y) \in S'$
13       $z := |B \cap S'|$ // The number of element in the intersection of the
       $S'$ and the subset of $[n] \times [m]$ that is represented by $C'$.
14       $loss_{\hat{C}} := 0$
15       **while** $z \geq 1$ **do**
16        **if** $u(a_i, b_i) \leq z$ **then**
17         $loss_{\hat{C}} := loss_{\hat{C}} + u(a_i, b_i) \cdot (\ell - b_i)^2$
18         $u(a_i, b_i) := 0$
19         $z := z - u(a_i, b_i)$
20         $i := i + 1$
21        **else**
22         $loss_{\hat{C}} := loss_{\hat{C}} + z \cdot (\ell - b_i)^2$
23         $u(a_i, b_i) := u(a_i, b_i) - z$
24         $z := 0$
25       $loss := loss + loss_{\hat{C}}$.
26 **return** $loss$

---

**Lemma 14.** *Let $D = \{(x_1, y_1), \cdots, (x_N, y_N)\}$ be an $n \times m$ signal i.e., $N := nm$. Let $k \geq 1$ be an integer, $\varepsilon \in (0, 1/4)$ be an error parameter, and $(C, u)$ be the output of a call to* SIGNAL-CORESET$(D, k, \varepsilon)$ *(see Algorithm 3). Let $s$ be a $k$-segmentation (in particular, a $k$-tree). Finally, let loss be the output of a call to* FITTING-LOSS$((C, u), s)$; *see Algorithm 5. Then there is a sufficiently large constant $\Delta \geq 1$ such that*

$$|\ell(D, s) - loss| \leq \Delta\varepsilon \cdot \ell(D, s).$$

*Moreover, loss can be computed in $O(k|C|)$ time.*

*Proof.* We consider the variables defined in Algorithm 5.

First, consider a subset $\hat{C}$ of $C$ from some iteration of the For loop at Line 2 of Algorithm 5, and let $B$ be the sub-signal that corresponds to $\hat{C}$, as in Line 3. We now prove that the loss $loss_{\hat{C}}$ computed in the same iteration of the For loop (i.e., at Lines 3- 25) satisfies the following claim.

**Claim 14.1.** *Let $z = |\{s(x) \mid (x, y) \in B\}|$ be the number of distinct values $s$ assigns to the coordinates of $B$ (as computed in Line 4). Then, $loss_{\hat{C}}$ satisfies that*

$$\begin{cases} \ell(B, s) = loss_{\hat{C}} & \text{if } z = 1 \\ |\ell(B, s) - loss_{\hat{C}}| \leq \varepsilon \cdot \ell(B, s) + O\left(\frac{\text{opt}_1(B)}{\varepsilon}\right) & \text{otherwise} \end{cases}$$

*Proof.* We prove Claim 14.1 using the following case analysis: (i) $z = 1$ and (ii) $z \geq 2$.

**Case (i):** $z = 1$. We prove that $\ell(B, s) = loss_{\hat{C}}$.

Since the input coreset $(C, u)$ was computed using Algorithm 3, we know that the set $\hat{C}$ was computed at Line 5 of Algorithm 3, along with a weight function $\hat{u}$. Hence, the pair $(\hat{C}, \hat{u})$ satisfy, by construction, the following property:

$$\sum_{(a,b) \in \hat{C}} \hat{u}((a, b)) \cdot (b \mid b^2 \mid 1) = \sum_{(x,y) \in B} (y \mid y^2 \mid 1). \tag{5}$$

Now, for any constant $\hat{s} \in \mathbb{R}$, we have that

$$\begin{aligned} &\sum_{(a,b) \in \hat{C}} \hat{u}(a, b)(b - \hat{s})^2 \\ &= \sum_{(a,b) \in \hat{C}} \hat{u}(a, b) \cdot b^2 + \hat{s}^2 \sum_{(a,b) \in \hat{C}} \hat{u}(a, b) - 2\hat{s} \sum_{(a,b) \in \hat{C}} \hat{u}(a, b)b \\ &= \sum_{(x,y) \in B} y^2 + \hat{s}^2 \sum_{(x,y) \in B} 1 - 2\hat{s} \sum_{(x,y) \in B} y \\ &= \sum_{(x,y) \in B} (y - \hat{s})^2, \end{aligned} \tag{6}$$

where the second equality is by (5).

Since $z = |\{s(x) \mid (x, y) \in B\}| = 1$, there is a constant $\hat{s} \in \mathbb{R}$ such that

$$\ell(B, s) = \sum_{(x,y) \in B} (y - \hat{s})^2. \tag{7}$$

Hence, we have that

$$\ell(B, s) = \sum_{(x,y) \in B} (y - \hat{s})^2 = \sum_{(a,b) \in \hat{C}} \hat{u}(a, b)(b - \hat{s})^2 = loss_{\hat{C}},$$

where the first derivation is by (7), the second derivation is by (6), and the last derivation is by the definition of $loss_{\hat{C}}$ at Line 7 of Algorithm 5.

**Case (ii):** $z \geq 2$. We prove that

$$\left| \ell(B, s) - loss_{\hat{C}} \right| \leq \varepsilon \cdot \ell(B, s) + O\left(\frac{\mathrm{opt}_1(B)}{\varepsilon}\right).$$

We first observe that, by the triangle inequality, for any $a, b, c \in \mathbb{R}$ we have that

$$
\begin{aligned}
\left| |a - c|^2 - |b - c|^2 \right| &= ||a - c| - |b - c|| \cdot (|a - c| + |b - c|) \\
&\leq |a - b| \cdot (2|a - c| + |a - b|) \\
&= |a - b|^2 + 2|a - c| \cdot |a - b| \\
&= |a - b|^2 + 2\sqrt{\varepsilon}|a - c| \cdot \frac{|a - b|}{\sqrt{\varepsilon}} \\
&\leq |a - b|^2 + \varepsilon \cdot |a - c|^2 + \frac{|a - b|^2}{\varepsilon} \\
&= \varepsilon \cdot |a - c|^2 + \left(1 + \frac{1}{\varepsilon}\right) \cdot (a - b)^2,
\end{aligned}
\tag{8}
$$

where the second inequality holds since $2xy \leq x^2 + y^2$ for every $x, y \in \mathbb{R}$.

**Smoothed coreset.** In Algorithm 3 we computed some small compression $C_B$, along with a weights function $u_B$, for every subset $B$ in the partition of the input. The size $|C_B|$ of this compression is a small constant, independent of the (potentially large) size of $B$. The pair $(C_B, u_B)$ satisfy a set of properties, which we visually demonstrate via this "smoothed coreset" notion; see Fig. 3. Informally, the "smoothed version" of $(C_B, u_B)$ is another pair $(C'_B, u'_B)$, such that $C'_B$ contains a duplication of the elements of $C_B$. The number of duplications of every element $c$ from $C_B$ is according to its weight $u_B(c)$.

We now formally define a "smoothed version" of a pair $(\hat{C}, \hat{u})$. A pair $(S, w)$ is said to be a *smoothed version* of the pair $(\hat{C}, \hat{u})$ if it satisfies the following properties: (i) $(S, w)$ has the same sum of weights, sum of labels, and sum of squared labels as $(\hat{C}, \hat{u})$, (ii) The set of coordinates $\{a | (a, b) \in S\}$ in $S$ covers the entire set of coordinates $\{x | (x, y) \in B\}$ of the original set $B$, with possible duplicates, and (iii) The sum of weights over all elements in $S$ with the same coordinate is 1. Formally,

$$\sum_{(a,b) \in S} w((a, b)) \cdot (b \mid b^2 \mid 1) = \sum_{(a,b) \in \hat{C}} \hat{u}((a, b))(b \mid b^2 \mid 1), \tag{9}$$

$$\{x | (x, y) \in B\} = \{a | (a, b) \in S\}, \tag{10}$$

and

$$\sum_{(a,b) \in S : a = x} w((a, b)) = 1 \text{ for every } (x, y) \in B. \tag{11}$$

In what follows, for every pair $(S, w)$ which is a smoothed version of $(\hat{C}, \hat{u})$, we prove the following two properties: We now prove the following two properties: first, that

$$\left| \ell(B, s) - \ell((S, w), s) \right| \leq \varepsilon \cdot \ell(B, s) + O\left(\frac{\mathrm{opt}_1(B)}{\varepsilon}\right), \tag{12}$$

for every every pair $(S, w)$ which is a smoothed version of $(\hat{C}, \hat{u})$. Second, we need to prove there is some pair $(\hat{S}, \hat{w})$ which is a smoothed version of $(\hat{C}, \hat{u})$ that satisfies

$$loss_{\hat{C}} = \ell((\hat{S}, \hat{w}), s), \tag{13}$$

where $loss_{\hat{C}}$ is the loss computed at Lines 3- 25, using only the pair $(\hat{C}, \hat{u})$ (i.e., at the current iteration of the outer-most For loop of Algorithm 5), without actually computing $(\hat{S}, \hat{w})$). Case (ii) then immediately holds by combining (12) and (13) above.

**A proof of** (12). Let $(S, w)$ be a pair which is a smoothed version of $(\hat{C}, \hat{u})$. By definition of $(S, w)$, we have that $w$ sums to 1 over all $(a, b) \in S$ with the same $a$, as in (11). Therefore, for every

$(x, y) \in B$ we can rewrite the term $(y - s(x))^2$ as $\sum_{(a,b) \in S: a=x} w(a, b)(y - s(x))^2$. Now, define $y_B(x) = y$ for every $(x, y) \in B$. We therefore have that

$$
\begin{aligned}
\ell(B, s) &= \sum_{(x,y) \in B} (y - s(x))^2 \\
&= \sum_{(x,y) \in B} \left( \sum_{(a,b) \in S: a=x} w(a, b) \right) \cdot (y - s(x))^2 \qquad (14) \\
&= \sum_{(x,y) \in S} w(x, y)(y_B(x) - s(x))^2,
\end{aligned}
$$

where the last equality holds by (10) and by simply combining the two sums.

We now have that

$$
\begin{aligned}
&|\ell(B, s) - \ell((S, u), s)| \\
&= \left| \sum_{(x,y) \in S} w(x, y)(y_B(x) - s(x))^2 - \sum_{(x,y) \in S} w(x, y)(y - s(x))^2 \right| \qquad (15) \\
&= \left| \sum_{(x,y) \in S} w(x, y) \cdot \left( (y_B(x) - s(x))^2 - (y - s(x))^2 \right) \right| \\
&\leq \sum_{(x,y) \in S} w(x, y) \left| (y_B(x) - s(x))^2 - (y - s(x))^2 \right| \qquad (16) \\
&\leq \sum_{(x,y) \in S} w(x, y) \left( \varepsilon \cdot (y_B(x) - s(x))^2 + \left( 1 + \frac{1}{\varepsilon} \right) (y_B(x) - y)^2 \right) \qquad (17) \\
&= \varepsilon \cdot \sum_{(x,y) \in S} u(x, y) \cdot (y_B(x) - s(x))^2 + \left( 1 + \frac{1}{\varepsilon} \right) \sum_{(x,y) \in S} w(x, y) \cdot (y_B(x) - y)^2 \\
&= \varepsilon \cdot \ell(B, s) + \left( 1 + \frac{1}{\varepsilon} \right) \sum_{(x,y) \in S} u(x, y) \cdot (y_B(x) - y)^2, \qquad (18)
\end{aligned}
$$

where (15) is by combining (14) and the definition of $\ell$, (16) holds since the sum of absolute values is greater or equal than the absolute value of a sum, (17) holds by substituting in (8) every term in the sum, and (18) is by (14).

We now bound the rightmost term of (18). Let $\hat{s} \equiv 1/|B| \sum_{(x,y) \in B} y$ be a 1-segmentation function that returns the label mean of $B$. We have that

$$
\begin{aligned}
&\sum_{(x,y) \in S} w(x, y) \cdot (y_B(x) - y)^2 \\
&\leq 2 \cdot \sum_{(x,y) \in S} w(x, y) \cdot \left( (y_B(x) - \hat{s}(x))^2 + (y - \hat{s}(x))^2 \right) \qquad (19) \\
&= 2 \sum_{(x,y) \in S} w(x, y) \cdot (y_B(x) - \hat{s}(x))^2 \\
&\quad + 2 \sum_{(x,y) \in S} w(x, y) \cdot (y - \hat{s}(x))^2 \\
&= 2 \cdot (\ell(B, \hat{s}) + \ell((S, u), \hat{s})) \qquad (20) \\
&= 2 \cdot (\ell(B, \hat{s}) + \ell(B, \hat{s})) \qquad (21) \\
&= 4 \cdot \ell(B, \hat{s}) \\
&= 4 \cdot \mathrm{opt}_1(B) \qquad (22)
\end{aligned}
$$

where (19) is by the weak triangle inequality, (20) is by combining the definition of $\ell$ with (14), (21) holds by Case (i) above, and (22) holds since the label means minimizes its sum of squared differences to the labels.

Figure 8: **(Left):** The pair $(\hat{C}, \hat{u})$. **(Middle):** A 4-segmentation $s$, which induces a partition of $[5] \times [5]$ into 4 sets $\mathcal{B} = B_1 \cup B_2 \cup B_3 \cup B_4$ where $B_1 = \{(1,1), (2,1), (3,1), (4,2), \cdots, \}$, $B_2 = \{(1,3), (2,3), (1,4), (2,4), \cdots, \}$, $B_3 = \{(3,3), (4,3), (5,3)\}$, $B_4 = \{(3,4), (4,4), (4,5), (3,5), \cdots\}$. **(Right):** A smoothed version $(\hat{S}, \hat{w})$ of $(\hat{C}, \hat{u})$. There can be more than one unique smoothed version for the same pair $(\hat{C}, \hat{u})$; see Properties in (9)-(11). The pair $(\hat{S}, \hat{w})$ is constructed by iterating over every set $B \in \mathcal{B}$. Every element in $B$ is assigned to a label from the labels of the 4-coreset points $(8, 17, 21, 22)$ as follows: If $|B| > \hat{u}(l_i)$, then $\hat{u}(l_i)$ elements of $B$ are assigned to $l_1$ and $|B| - \hat{u}(l_i)$ are assigned to $l_{i+1}$. If $|B| \le \hat{u}(l_i)$ then all the elements of $B$ are assigned to $\hat{u}(l_i)$, and $|B|$ is subtracted from $\hat{u}(l_i)$, and so on. If $\hat{u}$ assigns fractional weights, then some elements of $B$ might be assigned to more than one label, as long as the sum of weights over every element in $B$ is 1. By construction, $(\hat{S}, \hat{w})$ satisfies Properties (10)-(11). Hence, $(\hat{S}, \hat{w})$ is a smoothed version of $\hat{C}, \hat{u}$. Computing $\ell((\hat{S}, \hat{w}, s)$ can be trivially computed in time only $O(k|\hat{C}|)$ (rather than $O(n)$ where $n$ is the size of the original data), since the sets in the partition $\mathcal{B}$ contain a duplication of a constant number of labels.

Equation (12) now holds by combining (18) and (22).

**A proof of** (13). To prove (13), in Fig. 8 we construct a smoothed version $(\hat{S}, \hat{w})$ of $(\hat{C}, \hat{u})$ which satisfies (13).

Furthermore, by combining the construction of $(\hat{S}, \hat{w})$ with the computation of $loss_{\hat{C}}$ in Lines 9-25 of Algorithm 5, we obtain, as desired, that $loss_{\hat{C}} = \ell((\hat{S}, \hat{w}), s)$.

Claim 14.1 now holds by combining cases (i) and (ii) above.

$\square$

**We now prove Lemma 14.** Consider the construction of $(C, u)$ in Algorithm 3. By definition and by Lemma 5, the function $s'$ computed at Line 5 of Algorithm 3 is an $O(k^8 \log^2 nm)$-segmentation and satisfies that

$$\ell(D, s') \in O\left(k \log n \cdot \text{opt}_k(D)\right).$$

By the last derivation, let $c_\alpha$ be the smallest constant such that

$$\ell(D, s') \le c_\alpha \cdot k \log n \cdot \text{opt}_k(D).$$

By the last inequality and definitions of $\sigma$ and $\alpha$ we obtain that

$$
\begin{aligned}
\sigma := \frac{\ell(D, s')}{\alpha} &\le \frac{c_\alpha k \log nm \cdot \text{opt}_k(D)}{\alpha} \\
&= \frac{c_\alpha k \log nm \cdot \text{opt}_k(D)}{c_\alpha k \log nm} \\
&= \text{opt}_k(D).
\end{aligned}
\tag{23}
$$

Now consider the partition $\mathcal{B}$ computed at Line 3 of Algorithm 3 via a call to PARTITION$(D, \gamma, \sigma)$. By Lemma 7, $\mathcal{B}$ satisfies that

(i) $\text{opt}_1(B) \le \gamma^2 \sigma$ for every $B \in \mathcal{B}$.

(ii) $\mathcal{B}$ is a partition of $D$ whose size is $|\mathcal{B}| \in O\left(\frac{\alpha}{\gamma^2}\right)$.

(iii) There are $O\left(\frac{k\alpha}{\gamma}\right)$ sub-signals $B \in \mathcal{B}$ where $|\{s(x) \mid (x,y) \in B\}| > 1$.

(iv) $\mathcal{B}$ can be computed in $O(|D|)$ time.

Consider the pair $(C_B, u_B)$ computed at Line 5 of Algorithm 3 for some $B \in \mathcal{B}$. Now, consider the For loop at Line 2 of Algorithm 5. For every pair $(C_B, u_B)$, there is an iteration of this For loop for which $C' = C_B$. In this iteration, Algorithm 5 computes a loss $loss_{C_B}$ that corresponds to $(C_B, u_B)$. We can plug $B$, $(C_B, u_B)$, and $loss_{C_B}$ in Claim 14.1 to obtain that:

$$\begin{cases} \ell(B,s) = loss_{C_B} & \text{if z = 1} \\ |\ell(B,s) - loss_{C_B}| \leq \varepsilon \cdot \ell(B,s) + O\left(\frac{\text{opt}_1(B)}{\varepsilon}\right) & \text{otherwise} \end{cases} \tag{24}$$

Let $\mathcal{B}_1 \subseteq \mathcal{B}$ contain the set of sub-signals in $\mathcal{B}$ that are not intersected by $s$, i.e., $\mathcal{B}_1 = \{B \in \mathcal{B} \mid |s(B)| = 1|\}$, and let $\mathcal{B}_2 = \mathcal{B} \setminus \mathcal{B}_1$ be the set of sub-signals which are partially intersected by $s$.

By Property (iii) above,

$$|\mathcal{B}_2| \in O\left(\frac{k\alpha}{\gamma}\right). \tag{25}$$

Furthermore, by combining (24) with Property (i) above, for every $B \in \mathcal{B}_2$ we have that

$$|\ell(B,s) - loss_{C_B}| \leq \varepsilon \cdot \ell(B,s) + O\left(\frac{\text{opt}_1(B)}{\varepsilon}\right) \leq \varepsilon \cdot \ell(B,s) + O\left(\frac{\gamma^2 \sigma}{\varepsilon}\right). \tag{26}$$

In other words, the loss $\ell(B,s)$ of every sub-signal $B \in \mathcal{B}_2$ is approximated by $loss_{C_B}$ up to some small error. Hence, by summing over all $B \in \mathcal{B}_2$ we obtain that

$$\sum_{B \in \mathcal{B}_2} |\ell(B,s) - loss_{C_B}|$$

$$\in \sum_{B \in \mathcal{B}_2} \left(\varepsilon \cdot \ell(B,s) + O\left(\frac{\gamma^2 \sigma}{\varepsilon}\right)\right) \tag{27}$$

$$\leq \varepsilon \cdot \ell(D,s) + O\left(|\mathcal{B}_2| \cdot \frac{\gamma^2 \sigma}{\varepsilon}\right) \tag{28}$$

$$\leq \varepsilon \cdot \ell(D,s) + O\left(\frac{k\alpha}{\gamma} \cdot \frac{\gamma^2 \sigma}{\varepsilon}\right)$$

$$\leq \varepsilon \cdot \ell(D,s) + O\left(\frac{k\alpha\gamma}{\varepsilon} \cdot \text{opt}_k(D)\right) \tag{29}$$

$$\leq \varepsilon \cdot \ell(D,s) + O(\varepsilon \cdot \text{opt}_k(D)) \tag{30}$$

$$\in O(\varepsilon \cdot \ell(D,s)), \tag{31}$$

where (27) follows from (26), (28) is by (25), (29) is by (23), (30) holds since $k\alpha\gamma \leq \varepsilon^2$, and (31) holds since $\text{opt}_k(D) \leq \ell(D,s)$ for every $k$-segmentation $s$.

Furthermore, for every $B \in \mathcal{B}_1$, by (24) we have that $\ell(B,s) = loss_{C_B}$. Hence, by summing over every $B \in \mathcal{B}_1$ we obtain that

$$\sum_{B \in \mathcal{B}_1} \ell(B,s) = \sum_{B \in \mathcal{B}_1} loss_{C_B}. \tag{32}$$

In other words, the loss $\ell(B,s)$ of ever sub-signal $B \in \mathcal{B}_1$ is accurately estimated by $loss_{C_B}$.

Algorithm 5 then outputs the sum of losses

$$loss := \sum B \in \mathcal{B} loss_{C_B}. \tag{33}$$

We hence obtain that

$$|\ell(D,s) - loss| = \left|\sum_{B \in \mathcal{B}_1} (\ell(B,s) - loss_{C_B}) + \sum_{B \in \mathcal{B}_2} (\ell(B,s) - loss_{C_B})\right|$$

$$\leq \left|\sum_{B \in \mathcal{B}_1} (\ell(B,s) - loss_{C_B})\right| + \left|\sum_{B \in \mathcal{B}_2} (\ell(B,s) - loss_{C_B})\right| \in O(\varepsilon \cdot \ell(D,s)), \tag{34}$$

where the first derivation is by 33, second derivation is by the triangle inequality, and the last is by combining (32) and (31).

By (34), there is a constant $\Delta \geq 1$ such that
$$|\ell(D,s) - loss| \leq \Delta \varepsilon \ell(D,s).$$
This concludes the proof of the claim in Lemma 14.

**Computational time:** Line 2 of the Algorithm 5 is a loop with $\frac{|C|}{|\hat{C}|}$ iterations. Inside this loop: if $z > 1$ line 7 is computed in $O(|\hat{C}|)$, else line 11 is another loop with $O(k)$ iterations, inside which the line 20 is executed at most $|\hat{C}|$ times and line 24 can be executed only once because it results in $z = 0$ and in exiting from the while loop.

In total the complexity of line 15 is $O(|\hat{C}|)$, of line 11 is $O(k|\hat{C}|)$ and of line 2 and the whole algorithm: $O(k|C|)$

**Space complexity:** Algorithm 5 uses only constant amount of additional storage space because in each line of the algorithm only numeric variables are created and variables are reused inside the loops.

$\square$

**Theorem 15** (Coreset). *Let $D = \{(x_1, y_1), \cdots, (x_N, y_N)\}$ be an $n \times m$ signal i.e., $N := nm$. Let $k \geq 1$ be an integer (that corresponds to the number of leaves/rectangles), and $\varepsilon \in (0, 1/4)$ be an error parameter. Let $(C, u)$ be the output of a call to $\textsc{Signal-Coreset}(D, k, \varepsilon/\Delta)$ for a sufficiently large constant $\Delta \geq 1$; see Algorithm 3. Then, $(C, u)$ is a $(k, \varepsilon)$-coreset for $D$ of size $|C| \in \frac{(k \log(N))^{O(1)}}{\varepsilon^4}$; see Definition 3. Moreover, $(C, u)$ can be computed in $O(kN)$ time.*

*Proof.* To prove that $(C, u)$ is a $(k, \varepsilon)$-coreset for $D$, we need to prove that for every $k$-segmentation $s$, $(C, u)$ suffices to approximate the loss $\ell(D, s)$, up to a multiplicative factor of $1 + \varepsilon$, in time that depends only on $|C|$ and $k$.

Let $s$ be a $k$-segmentation and let $loss \geq 0$ be an output of a call to $\textsc{Fitting-loss}((C, u), s)$; see Algorithm 5. Then, by Lemma 14, $loss$ can be computed in $O(k|C|)$ time (i.e., in time that depends only on $|C|$ and $k$), and provides, as required, a $(1 + \varepsilon)$-approximation to $\ell(D, s)$ as
$$|\ell(D, s) - loss| \leq \Delta \cdot \frac{\varepsilon}{\Delta} \cdot \ell(D, s) = \varepsilon \cdot \ell(D, s).$$
Hence, $(C, u)$ is a $(k, \varepsilon)$-coreset for $D$.

Line 1 of Algorithm 3 can be computed in $O(k \cdot |D|)$ time by Lemma 5. Line 3 of Algorithm 3 can be computed in $O(|D|)$ time by Lemma 7. The loop at Line 4 can be computed in $\sum_{B \in \mathcal{B}} O(|B|) = O(|D|)$ time by Section E. Hence, the call $\textsc{Signal-Coreset}(D, k, \varepsilon)$ can be implemented in $O(k|D|) = O(kmn)$ time.

**Proof behind Line 6.** We now prove that the replacements of the coordinates applied at Line 6 does not violate the correctness of the algorithm. Observe that replacing the coordinates of entries inside each cell, while keeping the same labels, does not affect the variance of this subset. Therefore, the cost of this cell, which is computed in Algorithm 5) and depends only on the labels, remains exactly the same.

**Space complexity:** By construction, each pair $(C_B, u_B)$ computed at Line 5 can be stored using only $O(1)$ space. Hence, the concatenation $(C, u)$ of the $|B|$ pairs $\{(C_B, u_B) \mid B \in \mathcal{B}\}$ can be stored using $O(|\mathcal{B}|) \in O(\alpha/\gamma^2) = O(\alpha(\beta k)^2/\varepsilon^4) = O\left(\frac{k^{O(1)} \log^{O(1)} nm}{\varepsilon^4}\right)$ space. $\square$

# E   The Caratheodory Theorem

Given a point $p \in \mathbb{R}^d$ inside the convex hull of a set of points $P \subseteq \mathbb{R}^d$, Caratheodory's Theorem proves that there is a subset of at most $d + 1$ points in $P$ whose convex hull also contains $p$.

**Theorem 16** (Caratheodory's Theorem [13, 48]). *Let $P \subseteq \mathbb{R}^d$ be a (multi)set of $n$ points. Then in $O(nd^3)$ time we can compute a subset $Q \subseteq P$ and a weights functions $u : Q \to [0, \infty)$ such that: (i) $Q \subseteq P$, (ii) $|Q| = d + 1$, (iii) $\sum_{q \in Q} u(q) \cdot q = \frac{1}{n} \sum_{p \in P} p$, and (iv) $\sum_{q \in Q} u(q) = n$.*

**Corollary 17.** *Let $D$ be an $n \times m$ sub-signal. Then, in $O(|D|)$ time we can compute a weighted $n \times m$ sub-signal $(A, w)$ such that: (i) $A \subseteq D$, (ii) $|A| = 4$, (iii)* $\sum_{(a,b)\in A} w(a,b) \cdot (b \mid b^2 \mid 1) =$

$$\sum_{(x,y)\in D} (y \mid y^2 \mid 1), \text{ and } \sum_{(a,b)\in A} w(a,b) = |D|.$$

*Proof.* Define the multi-set $P = \{(y \mid y^2 \mid 1) \in \mathbb{R}^3 \mid (x,y) \in D\}$. Now, substituting $P$ in Theorem 16 yields that in $O(n)$ time we can compute a subset $Q \subseteq P$ and a weights functions $u : Q \to [0, \infty)$ such that: (i) $Q \subseteq P$, (ii) $|Q| = 4$, (iii) $\sum_{q\in Q} u(q) \cdot q = \frac{1}{|D|} \sum_{p\in P} p$, and (iv) $\sum_{q\in Q} u(q) = |D|$.

Now, add to $A$ a single element $(x,y) \in D$ for every $(y \mid y^2 \mid 1) \in Q$. In other words, for every element chosen for the set $Q$ by the Caratheodory theorem, add its corresponding element from $D$ to $A$. Furthermore, define $w(x,y) = |D| \cdot u((y \mid y^2 \mid 1))$ for every $(x,y) \in A$. Corollary 17 trivially holds for $(A, w)$. $\qquad\square$