# OpenReview forum: "Coresets for Decision Trees of Signals"
_NeurIPS.cc/2021/Conference — NeurIPS 2021 Spotlight_

### Official Review · Reviewer_nDd9 · 2021-07-05

**Rating:** 4
**Confidence:** 4

**Summary:**

The paper provides a new perspective on scalability for computing decision trees, and discusses how to build coresets for these objects.  The problem is posed as a data set over R^d x R, and the goal is to decompose R^d into rectangles with constant values so that the sum of squared errors from each rectangle to the points within it is minimized.

**Ethical Concerns:**

The authors did not engage on this topic.

**Limitations And Societal Impact:**

The authors did not engage on this topic.

**Main Review:**

The paper devises a greedy approach.  It greedy decomposes data into contiguous sets columns with, then roughly the same procedure is applied on each column set, dividing into contiguous sets of rows.  It recurses alternating on dimensions.  The algorithm and examples seem to be described in the d=2 case, but the discussion and experiments apply to larger d (15 and 18).

Given the multi-dimensional histogram found, (with small extension) standard decision tree (and even random forest) algorithms can be run, and can be much more efficient.  Experiments show only a small loss in quality (say 3%) with about a factor 10 reduction in space and time.

There are some size bounds on the coreset assuming nice properties of the data.  The polynomial terms  in terms of log N  (Where N is the data set size) and k (the number of leaves) is not specified, but is constant.


Overall, I like the idea for coresets for decision trees.  I can imagine a series of follow-on work to refine this work.  However, I have several concerns about this work itself:

 - the algorithmic description is not written in a way that is easy to understand.  It took several reads to (Hopefully correctly) decipher what I think is going on.  Also, it seems only described for 2 dimensions/features.

 - there is related work in databases that looks at quite similar problems, and the relation is not discussed.  Namely work in R-trees and in sketching multi-dimensional histograms (e.g., https://dl.acm.org/doi/abs/10.1145/564691.564741 or https://dl.acm.org/doi/pdf/10.1145/564691.564741).  Given that this paper seems to propose a fairly simple greedy method, it would be useful to compare with this fairly extensive line of work.

 - the paper claims a deep connection with partition trees, but does not really explain it in the main 9 pages.  I find this odd in that partition trees were mainly developed to deal with halfspace queries (of *any* orientation) and other objects which can be lifted to be halfspaces or composed of them.  There is a mostly different but parallel line of work that developed range queries techniques for axis-aligned queries, which seem to align with the axis-aligned rectangles used in decision trees and the proposed coresets.

 - The paper's presented analysis is for a restricted set of decision tree problems.  It should work for any decision tree given that structure holds.  However, then experimentally they apply them to random forests.  It is not completely clear that (or how) the coresets apply to random forests.  There are two concerns: the multiple queries and the correlation between classifiers.  The first concern should probably be simple using a union bound or something.  But the second one I understand less - that is, if a coreset is too course, does it prevent the random forest from reaching as a high a precision since the difference between categories has been smoothed out.
  On the other hand, even vanilla decision trees still have an important place in ML since they are very explainable and favored among scientists who want a procedural explanation.  (random forests do not have this advantage -- this may be slightly conflated in the introduction).


Overall, I definitely like this direction, and I strongly encourage the authors to continue pursuing this.  However, I feel the writing of the paper could be significantly improved, so am not convinced this paper is ready for publication in NeurIPS.

**Time Spent Reviewing:**

3

---

> ### Author Response · Authors · 2021-08-09
> **Response to the Official Review of Paper2279 by Reviewer nDd9**
>
> *Reviewer*: “Overall, I like the idea for coreset for decision trees”.
> - Reply #1: Thank you very much.
>
> *Reviewer*: The algorithmic description is not well written.
> - Reply #2: We tried to improve them. Please note that in addition to the pseudo-code we added an overview and figures that seem to help the other reviewers.
> Finally, the open code to be provided can be easily read by non-expert programmers and contains many comments with references to the pseudo code.
> Regarding d>2, see Reply #5 in the response to Reviewer ybbQ above.
>
> *Reviewer*: Suggested related work in databases.
> - Reply #3: Thank you for the references. They are indeed relevant, and will be discussed.
>
> *Reviewer*: The relation to partition trees is not explicit in the paper.
> - Reply #4: We tried to clarify it under page limitations.
> The main idea behind partition trees is to partition the input into a (relatively small) number of subsets, each containing a fraction of the input, such that each query (in this case, a rectangular shape) might intersect only a small fraction of those subsets.
> The number of points contained in non-intersected subsets can be easily computed, while the sum of points in intersected subsets require a more involved solution. The novelty in their work is how to achieve such a partition of the input.
> In our work, we devise an algorithm which achieves the above requirements, but where the query is a decision tree (and not a rectangular shape), and the cost function is the MSE and not the number of points.
> More precisely, we provide an algorithm which partitions the input into a (relatively small) number of subsets, each containing a constant fraction of the input, such that any query (in this case, a decision tree), might intersect only a small fraction of those subsets. We then compress every subset via another novel algorithm such that the cost (MSE, in this case) of points contained in non-intersected subsets can be easily computed, while the cost of points in intersected subsets require a more involved calculation.
> In both cases the constant depends on epsilon and k.
> Following this reviewer's comment we noticed that partition trees are used recursively for exact computations ("range trees").
> This idea should also work for our coresets and was added to the Future Work section.
>
>
> *Reviewer*: Transition from decision trees to random forests is not clear.
> - Reply #5: The application for forest is straightforward when the trees are computed independently, as done for example in SKlearn’s random forest implementation. This is since our coreset can approximate, individually, the cost of every possible decision tree on the full data. Hence, our coreset also  trivially approximates the sum of costs of multiple decision trees (or any linear combination of their costs).
> For other cases, when there is dependency between the trees, it is indeed less clear and variations of our coreset might be needed. The question posed by the reviewer is a very interesting theoretical question, which will require a more thorough investigation.
> We emphasize this limitation and suggest it for future work.

---

### Official Review · Reviewer_BfeE · 2021-07-15

**Rating:** 7
**Confidence:** 4

**Summary:**

This paper studies the coreset construction for the decision tree problem. In the 2D k-decision tree problem, the goal is to partition the input n x m matrix into k submatrices (with some restrictions) such that the summation of the cost over all submatrices is minimized. This paper gives the first coreset construction: it outputs a small subset of indices of the matrix with new weights such that the cost of any decision tree over the coreset is a (1+-eps)-approximation of the cost over the input matrix. The running time is O(knm) and the coreset size is polynomial in klog(nm)/eps.

**Limitations And Societal Impact:**

I did not find any potentially negative societal impact of the work

**Main Review:**

Pros:
-Decision tree is a fundamental problem in computer science. To the best of my knowledge, this paper gives the first coreset construction for the decision tree problem. Authors provided a novel partitioning scheme and connected it to the techniques in the theory of VC dimension. Such combination is non-trivial and I really like their partitioning algorithm which is simple and clever.
-The algorithm is deterministic.
-The experiments showed that the running time can be significantly improved.
-The paper is well-written in general.

Cons:
-There is a lack of discussion of previous work which uses the techniques from PAC learning and VC dimension to construct coresets for other problems. For example, Feldman and Langberg STOC'2011 gives a unified framework to construct coresets for k-clustering by reducing estimating the clustering objective to the range query problem in the field of VC dimension. Although the technical details are totally different, the high level structures of the algorithms are similar: both first require a good bicriteria approximation, then partition the data and finally handle each part separately.
-Although authors claim that the techniques can be easily extended to higher dimensions (tensors), there is no formal statements. I think the precise statements for coreset size and running time for higher dimension d would be really appreciated in most applications with feature dimension > 2.

Other comments:
-What is the best possible dependency of the coreset size on k? I guess it is easy to prove that Omega(k) is a lower bound. But is linear dependence on k achievable? If we allow randomization, is it achievable?
-Typo: Definition 6 (i): "|\mathcal{B}|\leq c_1 n m" -> "|\mathcal{B}|\leq c_1"

**Time Spent Reviewing:**

3 hours

---

> ### Author Response · Authors · 2021-08-09
> **Response to the Official Review of Paper2279 by Reviewer BfeE**
>
> *Reviewer*: “I really like their partitioning algorithm”, “The paper is very-well written in general”.
> - Reply #1: We thank the reviewer for taking time to appreciate our algorithms.
>
> *Reviewer*: Missing discussion related to other coreset construction techniques.
> - Reply #2: Agreed. A general discussion about coreset construction frameworks, including citations, is added.
>
> *Reviewer*: No formal statement for d>2.
> - Reply #3: Correct. Due to lack of space and time we left the formal statement in higher dimensions for future work. However, following the reviewer’s recommendation, we will try to squeeze such a statement into this work.
>
> *Reviewer*: What is the lower bound for the dependency on k in the coreset size?
> - Reply #4: A linear bound in k is trivial and known for d=1 [47]. We suggest a polynomial dependency. Closing this gap is indeed an interesting problem that we added to the future work section.
>
> *Reviewer*: Typo.
> - Reply #5: Fixed.

---

### Official Review · Reviewer_ybbQ · 2021-07-16

**Rating:** 7
**Confidence:** 4

**Summary:**

The paper introduces small data summaries known as coresets for decision trees on a two dimensional data domain. It does so by summarizing rectangular blocks of data with their mean value as an approximation for each cell's individual value, hereby approximating up to (1+-eps) relative error the squared loss induced by each k-partition that can be formed by a decision tree with k leaves.
The construction is done in linear time in the input parameters and data and the size of the summary is poly(k log(n)/eps).
Some experiments show that the coresets are useful in practical settings where decision tree heuristics are compared on the coreset vs the original data.


**Ethical Concerns:**

-

**Limitations And Societal Impact:**

-

**Main Review:**

The paper reads very well and the work is well-motivated. In the introduction to coresets maybeit should be added that C is much smaller than the data D (right in lines 66-69).

it is just the squared error, not the MSE (l. 104)

The paper says there is no coresets in general, even in 1 dimension. but then for m x n-signals a coreset construction is presented. I find the discussion too vague. I didn't learn what was the problem that implies impossibility. And more importantly, what exactly is the relaxation that makes it possible again?
Can you explain? (and modify the paper accordingly)

Is an extension to d>2 tensor data even possible? Why is this only an outlook for future work and not dealt with? This is currently also vague (but minor).

since the coreset constructions are claimed to be different from existing sampling and greedy optimization, shouldn't there be some references on the appropriate surveys with more diverse coresets methods instead?

I found it weird that we have mxn signals and then nxm sub-signals. maybe the authors could use some other letters to denote the input dimenions?

Definition of the O-notation should have c>0 and x_0 such that the inequality holds for all x>x_0. (199-200)

t is overloaded  on page 6 before Sec. 3

Algo 2 inner while loop looks like the last iteration is run, but the result is not considered in the output. Please double check.

Fig 3  It seems that in the lower right corner there should be a low constant value and the lower left high value should be removed since it is covered by the other high value (22 and 21).

regarding those tuples (b|b^2|1) it seems very similar to k-means "clustering features" used for example in BIRCH. Another reference that should be cited in this context is https://dl.acm.org/doi/10.1145/1132863.1132873 which stores the sum of values, sum of squared values and number of elements, i.e., the sum of those tuples in your paper.

generally I suggest strongly to browse sudipto guha's DBLP for "histogram". There are a lot of extensions to other loss functions, fitting polynomials instead of constant values per bucket, streaming, more than one dimension. This seems very similar and strongly related to the contents of this submission or possible extensions.

alpha, beta >0  -> alpha, beta >1 (eg in Def4 and Lem7)

Coreset size: the authors claim that usually the theoretical bounds are much worse than in practice. Is this because the analysis is sloppy or loose? Are there any lower bounds? Or is this maybe since many "real-world" data sets from the usual ML repositories are easy to approximate?

**Time Spent Reviewing:**

6

---

> ### Author Response · Authors · 2021-08-09
> **Response to Official Review of Paper2279 by Reviewer ybbQ**
>
> *Reviewer*: “The paper reads very well and is well-motivated”.
> - Reply #1: We thank the reviewer for appreciating our work.
>
> *Reviewer*: Add that C is much smaller than the input.
> - Reply #2: Good idea. Done.
>
> *Reviewer*: Squared error and not MSE.
> - Reply #3: Fixed.
>
> *Reviewer*: The paper says that there is no coreset even for 1 dimension. So how come it suggests a coreset construction for m x n-signals? Can you explain this relaxation?
> - Reply #4: Indeed, there is no coreset of size smaller than n for an arbitrary set of n vectors, with an example input even for d=1 (n real numbers).
> Intuition: we can summarize a sequence 1,2,3,..n by taking the first and last numbers, but we cannot do this for an arbitrary set of n numbers.
> This justifies our assumption that the input is a signal (i.e., assigns a value for every coordinate in the domain).
> We then prove a coreset construction is possible for any 2-dimensional SIGNAL. The generalization for d-dimensional signals seems straightforward.
> Only in the experiments, this assumption turns into a relaxation, where the input is not necessarily a signal. Even in this case, the coreset had successfully approximately the full data, with very small error.
> This is mentioned already in the title of the paper, but we added a paragraph that emphasizes this fact.
>
> *Reviewer*: Is the extension to d>2 tensor data even possible?
> - Reply #5: It seems to us a straightforward generalization, which is why we choose d=2 for simplicity and due to space limitations.  Following this reviewer's comment , we added an intuition and summarized it as follows.
> For d=2, our balanced partition algorithm partitioned the data in the first dimension (into horizontal chunks), such that the number of inner partitions of each chunk, in the second dimension, is smaller than some number \gamma. This number of inner chunks is obtained by applying the 1-dimensional partitioning algorithm.
> For d=3, we will apply the above idea recursively on the first two dimensions, as follows:
> We will partition the first dimension into chunks, such that the number of inner partitions of each chunk is smaller than some number \gamma. The number of inner partitions of each chunk is now obtained by applying the above algorithm for d=2.
> For higher d values, we simply apply the algorithm above recursively on the first d-1 dimensions to obtain the number of “inner partitions”. The 1-dimensions partitioning algorithm is always invoked when reaching the last dimension.
>
> *Reviewer*: Missing discussion related to other coreset construction techniques.
> - Reply #6: Agreed. It is hard to summarize the numerous techniques, but we tried adding more references and text (including sampling and greedy ones).
>
> *Reviewer*: Confusing use of n and m.
> - Reply #7: This will be made clearer.
>
> *Reviewer*: Definition of O should be modified.
> - Reply #8: Thank you, we will refer to this alternative and common definition.
>
> *Reviewer*: t is overloaded on page 6.
> - Reply #9: Nice catch! Fixed.
>
> *Reviewer*: Problem with Fig. 3
> - Reply #10: We do not seem to understand the problem in Fig. (3).
>
> *Reviewer*: The use of the first three moments is similar in BIRCH and other references.
> - Reply #11: We first thank the reviewer for the suggestions.
> The fact that we can compute the sum of squared distances to arbitrary query points via the first three moments is indeed very old and very common.
> Any epsilon-coreset for the so-called 1-mean problem may be used.
> In one of his works, Sudipto et al. tackled the k-segmentation problem for d=1. However, the work did not provide a coreset, but rather a solution (which can be applied on a coreset).
> More relevant references will be added.
>
> *Reviewer*: Suggestions for useful papers that might help extend our work.
> - Reply #12: We thank the reviewer for the suggestions. We will try to utilize techniques from those papers for future extensions.
>
> *Reviewer*: Alpha, beta > 1.
> - Reply #13: Done.
>
> *Reviewer*: Why do the authors claim the coreset sizes in practice are much smaller than in theory?
> - Reply #14: This phenomenon is well known for coresets; see discussion e.g., in:
> Feldman, Dan. "Core-sets: Updated survey." Sampling Techniques for Supervised or Unsupervised Tasks (2020): 23-44.
> We list here a few reasons: worst-case artificial examples vs. average behaviour on structured real-world data, noise removing/smoothing by coresets, the fact that in practice we run heuristics that output a local minima (and not optimal solutions with global minimum), non-tight analysis (especially when it comes to constants).

---

> > ### Comment · Reviewer_ybbQ · 2021-09-03
> > **Thanks**
> >
> > Thanks for the clarifications. I will leave my rating unchanged. Please address all issues carefully when revising the manuscript.

---

### Official Review · Reviewer_1UMR · 2021-07-22

**Rating:** 8
**Confidence:** 2

**Summary:**

The paper presents an efficient construction of coresets for decision trees. For general decision trees, coresets are useless because they cannot be very small. However, the result in this paper focuses on decisions trees for a restricted form of signal, namely, a function on the discrete grid of points $\{1,2,\dots, n\} \times \{1,2,\dots, m\}$. The result is derived by relating this problem to the problem of constructing balanced partition trees for planar point sets. The theoretical result is also evaluated on synthetic and real datasets, showing impressive gains in runtime and storage requirements.

**Main Review:**

I find the result interesting, both from the perspective of discovering a useful subset of decision trees that can be shown to have coresets, and from the perspective of the elegant solution relating the problem to a geometric one. The result is perhaps not surprising, but useful.

**Time Spent Reviewing:**

4 hours

---

> ### Author Response · Authors · 2021-08-09
> **Response to Official Review of Paper2279 by Reviewer 1UMR**
>
> We thank the reviewer for appreciating the usefulness of our result.

---

### Decision · Program_Chairs · 2021-09-28

**Decision:**

Accept (Spotlight)

**Comment:**

The paper was considered to have technical novelty and strength as "absolutely above the bar" of NeurIPS. However, there were very substantial concerns about the writing quality of the paper, almost of the level of being considered unpublishable in the current form. The authors are highly encourage to polish the paper and address all the writing concerns from the reviews (including comparison to past work).

**Consistency Experiment:**

NeurIPS has a long history of experimentation. In 2014, NeurIPS ran an experiment in which 10% of submissions were reviewed by two independent committees to quantify the randomness in the review process. This year, we repeated a variant of this experiment to see how the quality of the review process has changed over time.  This paper was part of the experiment and was therefore assigned to two committees (consisting of reviewers, an Area Chair, and a Senior Area Chair) that reached independent decisions.  If both committees made the same recommendation, this recommendation was followed. If a single committee recommended acceptance, the paper was accepted (with the exception of a few cases in which the other committee identified what we considered a fatal flaw, e.g., an error in a key result).

This copy’s committee reached the following decision: **Accept (Oral)**

The other committee assigned to the paper recommended **Accept (Poster)**.  You can find the other set of reviews, along with any follow up discussion with the authors here:
https://openreview.net/forum?id=GvU4RvMwlGo